# Quantum reference frames for an indefinite metric

Anne-Catherine de la Hamette [1,2,5✉], Viktoria Kabel [1,2,5], Esteban Castro-Ruiz[3,4] & Časlav Brukner[1,2]

The current theories of quantum physics and general relativity on their own do not allow us to study situations in which the gravitational source is quantum. Here, we propose a strategy to determine the dynamics of objects in the presence of mass configurations in superposition, and hence an indefinite spacetime metric, using quantum reference frame (QRF) transformations. Specifically, we show that, as long as the mass configurations in the different branches are related via relative-distance-preserving transformations, one can use an extension of the current framework of QRFs to change to a frame in which the mass configuration becomes definite. Assuming covariance of dynamical laws under quantum coordinate transformations, this allows to use known physics to determine the dynamics. We apply this procedure to find the motion of a probe particle and the behavior of clocks near the mass configuration, and thus find the time dilation caused by a gravitating object in superposition. Comparison with other models shows that semi-classical gravity and gravitational collapse models do not obey the covariance of dynamical laws under quantum coordinate transformations.

[1] Vienna Center for Quantum Science and Technology (VCQ), Faculty of Physics, University of Vienna, Boltzmanngasse 5, A-1090 Vienna, Austria. [2] Institute for Quantum Optics and Quantum Information (IQOQI), Austrian Academy of Sciences, Boltzmanngasse 3, A-1090 Vienna, Austria. [3] Institute for Theoretical Physics, ETH Zurich, Zurich, Switzerland. [4] Université Paris-Saclay, Inria, CNRS, LMF, 91190 Gif-sur-Yvette, France. [5] These authors contributed equally: Anne-Catherine de la Hamette, Viktoria Kabel. ✉email: acdelahamette@gmail.com

The theory of general relativity provides us with excellent tools to study and predict the dynamics of objects in the vicinity of gravitating bodies, as long as the latter are considered to be classical. The theory of quantum physics describes the behavior of quantum systems with impressive precision, allowing us to study quantum phenomena such as interference, superposition, and entanglement. Moreover, quantum field theory in curved spacetime describes quantum fields in a classical, possibly curved, spacetime background. However, up until today, we are still lacking a theory that permits us to describe the dynamics of physical systems in the neighborhood of gravitational sources that are genuinely quantum in full generality. One might think that these issues only concern highly energetic or cosmological scenarios, but in fact they are inherent to the low energy regime as well. This was first noted by Richard Feynman at the 1957 Chapel Hill Conference, where he suggested that "one should think about designing an experiment which uses a gravitational link and at the same time shows quantum interference"[1,2]. The situation we consider here is precisely of this type: Given a configuration of masses in a spatial superposition, how do physical systems evolve in its vicinity?

Most full quantum gravity approaches such as string theory[3] and loop quantum gravity[4] have not attempted to answer this specific question for now as this would require a rigorous transition to the low energy regime. On the other hand, there are perturbative approaches to quantum gravity, such as linearized quantum gravity[5] or effective quantum descriptions of the gravitational quantum potential[6], which are able to make predictions but are based on perturbations on a fixed spacetime background and are limited to weak gravitational fields. Other models such as semi-classical gravity[7] or gravitational collapse models[8–13] can treat the aforementioned situations in more generality and predict an effectively classical gravitational field, at least after a short amount of time. In contrast, recent proposals argue for the quantum nature of the gravitational field and in particular predict the generation of gravity-induced entanglement and other quantum phenomena[14–20]. With rapid advances in measuring the gravitational field of microscopic source masses[21] as well as creating and verifying superpositions of large molecules[22], experimental tests which can corroborate or exclude some of these approaches are getting within reach[23]. The hope is that in the near future, these two states of the art can be combined to design experiments in which superpositions of gravitational quantum sources can be created and the nature of their gravitational field investigated. In the meantime, further insight relies on theoretical investigation and a proper conceptual understanding of these problems. We show here that the above-mentioned question can be answered without making any a priori assumptions about the nature of the gravitational field sourced by a mass configuration in superposition. The aim of this paper is to construct a rigorous argument that is based on a minimal set of assumptions, in particular an extended symmetry principle[24], and still allows us to predict the dynamical behavior of probe systems in the vicinity of masses in superposition. As the results we find are in line with gravitational fields in superposition, they provide an independent justification for developing a quantum formalism for the spacetime metric. Conversely, if particular predictions based on the assumption that a quantum object sources a gravitational field in superposition are experimentally verified in the future, this would provide empirical evidence for our extended symmetry principle.

The main tool used in the construction of this argument are quantum reference frames (QRFs). The approach of QRFs has received a lot of attention in recent years, from the quantum gravity community[25–27] and the quantum information and quantum foundations community alike[28–52]. Most approaches to quantum reference frames take as a premise that the reference system relative to which a physical system is described needs to be explicitly included into the description. While several different frameworks can be found in the literature, we will make use of a combination of specific formalisms[35,40]. We refer the reader to Supplementary Note 1 for a short introduction to quantum reference frames.

Our argument is based on an extended symmetry principle: the "covariance of dynamical laws under quantum coordinate transformations". This assumption allows us to change from a situation in which a massive object is in superposition of spatial locations to a situation in which it is definite, that is, in a single position, and thus the gravitational field and metric are well-defined. We can then determine the time evolution in this frame and use the inverse QRF transformation to obtain the dynamics in the original frame. Just as with classical coordinate transformations, this is only possible if the dynamical laws are covariant under the reference frame transformations. In general relativity, the dynamical evolution is governed by the Einstein field equations. For completeness, we remind the reader of their explicit form:

$$R_{\mu\nu} - \frac{1}{2}Rg_{\mu\nu} + \Lambda g_{\mu\nu} = \frac{8\pi G}{c^4}T_{\mu\nu}, \tag{1}$$

where $R_{\mu\nu}$ is the Ricci curvature tensor, $R$ the scalar curvature, $g_{\mu\nu}$ the metric tensor, $T_{\mu\nu}$ the stress-energy tensor, $\Lambda$ is the cosmological constant, $G$ the Newtonian constant of gravitation, and $c$ the speed of light in vacuum. Note that the Einstein field equations are covariant under general classical coordinate transformations, which change the Einstein tensor $G_{\mu\nu} = R_{\mu\nu} - \frac{1}{2}Rg_{\mu\nu}$ and the energy-momentum tensor $T_{\mu\nu}$ in the same manner. The goal of this paper is to find an extension of these to "quantum coordinate transformations", i.e. superpositions of the classical coordinate transformations, and subsequently use them to solve the aforementioned typical problems in the low-energy quantum gravity regime. In general, this seems to require the introduction of a quantum state and corresponding Hilbert space for the metric tensor as it transforms non-trivially under general coordinate transformations. However, this additional step can be avoided if the metric tensor remains invariant under the applied coordinate transformations, i.e. if they are isometries. We can further consider transformations that move all systems, including the gravitational source, rigidly. By this, we mean that the relative coordinate distances between all systems remain invariant, where the coordinates are defined by a physically instantiated reference system (see "The case of one massive object" in "Results" section for a more detailed discussion). These additional transformations, which amount to global translations and rotations of all physical systems, do not affect the dynamics. If we want to be able to change into a frame in which the mass configuration is definite by means of these transformations and at the same time preserve the dynamical laws, we have to restrict to situations in which the mass configurations themselves are related by relative-distance-preserving transformations in the above sense. Before you put down this paper too quickly, note that this restriction does not imply that we can only consider trivial situations, in which the gravitational field is the same in each branch from the start. This would be true only if the gravitational source was the only object in the universe[53]. The presence of test particles breaks the equivalence of the mass configurations related by these transformations in different branches. In particular, we can still consider situations in which the distance between the mass configuration and the test particle is in a superposition. Given a situation in which mass configurations are related by relative-distance-preserving transformations across different branches, we can transform to the quantum frame in which the gravitational source becomes definite to compute the dynamical evolution of quantum

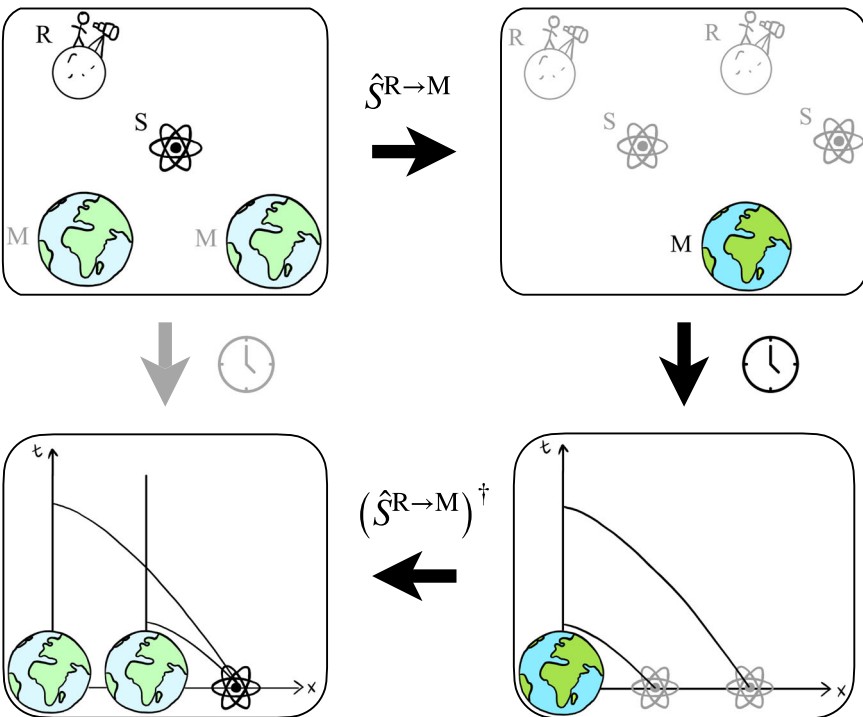

**Fig. 1 Qualitative depiction of the general argument in four steps.** We start in the frame of the reference system R with a probe system S and a mass configuration M that is in a quantum superposition (depicted by fainter colors). We apply the quantum reference frame transformation $\hat{S}^{R \to M}$ to move to the frame in which M and thus the spacetime metric are definite whereas the reference system R and the probe S are both in a superposition. In this frame, we can use known physics to solve for the dynamics of the probe system S. Here depicted is the geodesic motion of the probe S in the gravitational field sourced by a single mass M. Finally, we apply the inverse transformation $\left(\hat{S}^{R \to M}\right)^{\dagger}$ to the evolved state to change back to the original frame. We thus find the dynamics of S in the presence of an indefinite mass configuration. In particular, we observe that S moves in a superposition of trajectories and becomes entangled with the mass configuration M. All in all, the time evolution in the frame of R (gray arrow) is obtained by the above sequence of quantum reference frame changes and the time evolution in the frame of M (black arrows). Note that our argument is general and can be applied to cases beyond a single mass in superposition.

objects in its presence. Transforming back to the original frame, we are thus able to make concrete predictions for the motion of test particles or clocks in the presence of a gravitational source in superposition.

The general idea employed here is to use the relativity of superpositions[24] under QRF transformations to make an indefinite metric definite. This aspect is similar to the one in previous works on QRFs for indefinite metrics[43,49], however, there are important differences between the approaches. In recent work[43,49], a local quantum inertial frame is introduced, which can be associated with a frame attached to a quantum particle falling freely in a superposition of gravitational fields. It was shown that in the transition to the local quantum inertial frame, the metric becomes Minkowskian, extending the validity of Einstein's equivalence principle to quantum reference frames and spacetimes in general superpositions. However, it permits the transformation into a definite metric only in the infinitesimal vicinity of the origin of the local quantum inertial frame, whereas outside this range the metric remains indefinite. Our approach enables us to make the spacetime globally definite and is thus restricted to a superposition of configurations related by relative-distance-preserving transformations. Moreover, the above-mentioned previous works assume a Hilbert space structure for gravitational fields and thereby the assignment of a quantum state to the metric. Here, we avoid this methodological step.

## Results

**The general argument and its applicability.** We begin by stating the general argument, illustrated in Fig. 1. Consider a mass configuration M in superposition of semi-classical states with respect to a reference system R. Within this set-up, we would like to predict the motion of a sufficiently light probe S in the presence of the superposition. While the current established theories – quantum theory and general relativity separately – do not allow us to determine the gravitational field sourced by massive objects in superposition, we can solve the problem by transforming into a better suited quantum frame of reference. We achieve this by minimal assumptions and in a limit in which a future theory of quantum gravity is expected to have the same qualitative predictions. By applying a QRF transformation in the form of controlled shifts and rotations, depending on the position of M, we can change into the frame in which the mass is in a definite position while the test particle and the reference frame are in spatial superposition (Fig. 1). We now make the following assumption: Covariance of dynamical laws under quantum coordinate transformations: physical laws retain their form under quantum coordinate transformations. This assumption can be seen as a generalized symmetry principle, extending covariance under classical coordinate transformations to quantum superpositions thereof. It allows us to solve the dynamics in the reference frame of the mass configuration M and then transform back into the original frame to find the motion of the particle S in the presence of a mass in superposition. By changing to the reference frame of M, we enter the regime of a quantum particle S in the presence of a classical gravitational source M. In the most general case, the behavior of quantum objects in a fixed gravitational field is governed by quantum field theory on curved spacetime[54]. The regime has been studied both theoretically and,

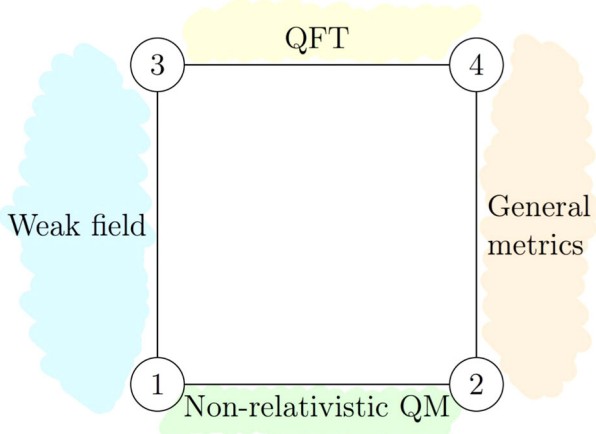

**Fig. 2 Discussion of different regimes of applicability.** The four different regimes characterized by the nature of the probe (vertical axis) and the gravitational field (horizontal axis). (1) Non-relativistic quantum mechanics (QM). Quantum particles in Newtonian gravity (regime covered by near-future table-top experiments). (2) Quantum particles on general spacetimes, among them quantum particles following semi-classical trajectories along the geodesics of general spacetimes. (3) Quantum field theory (QFT) in weak gravity. (4) Quantum field theory on curved spacetime. While our explicit calculations and examples mainly pertain to regime (1), our argument extends to regime (2) using the generalized operator introduced in "Generalization to *N* masses" subsection in the "Results" section. With an extension of the quantum reference frame change operator to treat probes as quantum fields, the same argument of using quantum reference frame transformations to change into a frame with definite metric also applies to regimes (3) and (4).

in the low energy limit, experimentally. In an interferometric experiment with neutrons in 1975, Colella, Overhauser, and Werner[55] observed a phase shift induced by the gravitational field of the Earth, establishing that different branches of a spatial superposition "feel" different gravitational potentials. Their results were improved and extended to atomic fountains to measure the phase shift due to Earth's gravitational potential[56], and to more general gravitational sources leading to more general spacetime curvatures[57]. While these works are restricted to a Newtonian gravitational field, an extension to general spacetimes was studied theoretically by Stodolsky[58]. He determined the motion of quantum particles along semi-classical paths in a general gravitational field and, in particular, calculated the quantum phase accumulated along these paths. Note that the different regimes regarding quantum systems on curved space-time are characterized by two limits: one regarding the quantum nature, from particles to fields, of the object experiencing the gravitational field and one regarding the strength of the gravitational field (Fig. 2).

The general argument presented in this article is in principle not restricted by either of these limits. Whenever a treatment of a situation with a definite spacetime is possible, it can be extended to the corresponding situation in which the gravitational source is in a quantum superposition of semi-classical states. Our explicit calculations and examples in the present section mainly focus on the regime of quantum particles in Newtonian gravity (1), which is also the regime of potential table-top experiments[23]. Note that the regime of quantum mechanics can be further subdivided into a realm in which only superpositions of semi-classical trajectories of quantum particles are considered and one in which particles follow a more general quantum evolution. In order to provide an intuition for the motion of the probe, we focus our attention in this section on the regime of quantum particles following semi-

classical trajectories. This means assuming that the gravitational fields vary sufficiently slowly within distances of the order of the De-Broglie wavelength of the probe for each semi-classical trajectory of the superposition and that the position and momentum associated to each amplitude are approximately well-defined at the same time[58]. In the "Methods" section, we will show how to make use of the Hamiltonian formulation of time evolution, which is available for instance in the Newtonian limit (see "Transformation of the Hamiltonian operator" section). This allows us to treat general quantum states for the probe. Quantum particles in the semi-classical subsector of regime (2) can be tackled with a generalized QRF transformation operator, which we introduce in the "Relative-distance-preserving transformation using an auxiliary system" subsection of the "Methods" section. Finally, the quantum field theoretic regimes (3) and (4), could be treated with an extension of our QRF change operator to act on probe systems which are quantum fields.

**The case of one massive object**. Let us now go through the steps of the argument in detail for the simple case of one massive object M in superposition of two well-defined, fixed, and classically distinguishable positions with respect to a quantum reference system R. By this, we mean that the mass is prepared and kept in a superposition of coherent states $|\mathbf{x}^{(i)}, \mathbf{p} = 0\rangle$ centered around mean positions $\mathbf{x}^{(i)}$ and zero momentum. This is motivated by most experimental proposals which involve masses that are kept at their initial position using optical or magnetic traps[23]. With strong potentials holding the mass in place, its time evolution becomes trivial. Moreover, due to the negligible magnitude of the momentum fluctuations in the coherent states, we neglect their contribution to the stress energy tensor. Finally, we take the mean positions $\mathbf{x}^{(i)}$ to be sufficiently far apart such that the overlap between the corresponding coherent states is negligible. This allows us to neglect the spread of the wavefunction in position space and effectively describe the massive object in terms of a superposition of position eigenstates $|\mathbf{x}^{(i)}\rangle$. This remains true throughout the duration of the experiment as the state of the massive object does not change over time. Note further that any issues regarding singularities that arise from treating a point mass could be dealt with by taking into account the extension of the massive object.

For all states $|\mathbf{x}\rangle$ in this article, $\mathbf{x}$ always refers to the coordinate distance between the system and the designated reference system and is associated with a concrete preparation procedure. We want to briefly address the question of how to assign well-defined coordinates in a frame in which the spacetime metric is indefinite. One option is motivated by the experimental procedure used to set up the situation: before the mass is put into a superposition, the spacetime metric is definite and taken to be Minkowski, which allows to assign coordinates to events operationally in a straightforward way. In particular, we can choose to assign the origin to the reference system and define the spatial coordinates of all other objects in the set-up through the Euclidean distances from the reference system. The spatial coordinates assigned to the massive object in superposition are then set by the positions of the traps used to hold the mass in this coordinate system, before the mass is placed in the traps. Even after inserting the massive object into the set-up, the coordinates remain well-defined. Furthermore, one can always give physical meaning to the coordinate distances in the frame of R, even after the massive object has been inserted, by finding the corresponding physical distances in the frame of M. More specifically, by changing into the frame of M in which the spacetime metric is definite, one can relate the transformed coordinates to the proper distances between objects in a well-defined spacetime.

To keep the reference system decoupled from the gravitational influence of the massive objects, it is usually assumed that it is infinitely far away from them. However, in the quantum mechanical treatment of the problem, one can achieve this decoupling operationally with less stringent requirements and, in particular, without imposing any further conditions on the configurations of the rest of the systems. This is detailed in Supplementary Note 2. A sufficiently light quantum probe S is initially placed in a definite position with respect to R, close enough to feel the gravitational pull of the mass M (this is illustrated in the top left subfigure of Fig. 1). We assume that the mass of the probe particle is small enough to neglect the gravitational field sourced by it. Moreover, for illustrative purposes, we treat the dynamics of the probe S in the semi-classical approximation in this section. In the "Transformation of the Hamiltonian operator" section, we show how to use the Hamiltonian formulation to determine the time evolution of a probe in a general quantum state.

*Reference frame of R.* Consider the joint state of R, M, and S in the reference frame of R,

$$|\psi\rangle_{\text{RMS}}^{(\text{R})} = |\mathbf{0}\rangle_{\text{R}} \frac{1}{\sqrt{2}} \left( \left|\mathbf{x}_{\text{M}}^{(1)}\right\rangle_{\text{M}} + \left|\mathbf{x}_{\text{M}}^{(2)}\right\rangle_{\text{M}} \right) |\mathbf{x}_{\text{S}}\rangle_{\text{S}}, \quad (2)$$

where $|\mathbf{x}\rangle$ denotes the eigenstate of the position operator relative to R while the superscripts (1) and (2) label the different branches. On the left-hand side, the superscript (R) indicates the reference frame and the subscript the systems described by the quantum state. Finally, note that we are following the notation of de la Hamette and Galley[40], in which the trivial state of the reference frame is included in the description.

*Reference frame of M.* To go into the reference frame of M, we perform a QRF transformation in the form of a controlled shift $\hat{U}_T = \hat{\mathcal{P}}_{\text{MR}} e^{\frac{i}{\hbar}\hat{x}_{\text{M}}\hat{p}_{\text{S}}}$[35], where the parity swap operator $\hat{\mathcal{P}}_{\text{MR}} \equiv \text{SWAP}_{\text{MR}} \circ \int dx |{-}x\rangle\langle x|_{\text{M}}$ exchanges the labels of M and R, taking into account relevant sign changes. The resulting state is

$$|\psi\rangle_{\text{MRS}}^{(\text{M})} = |\mathbf{0}\rangle_{\text{M}} \frac{1}{\sqrt{2}} \left( \left|-\mathbf{x}_{\text{M}}^{(1)}\right\rangle_{\text{R}} \left|\mathbf{x}_{\text{S}} - \mathbf{x}_{\text{M}}^{(1)}\right\rangle_{\text{S}} + \left|-\mathbf{x}_{\text{M}}^{(2)}\right\rangle_{\text{R}} \left|\mathbf{x}_{\text{S}} - \mathbf{x}_{\text{M}}^{(2)}\right\rangle_{\text{S}} \right), \quad (3)$$

describing the position of the particle S and the reference system R with respect to M.

*Dynamical evolution in the reference frame of M.* Since M is now in a definite position at the origin, we can determine the particle's motion. In the semi-classical approximation and assuming that no other forces act on the quantum particle or the massive object, which is taken to remain static with respect to R, the time evolution of the probe is governed up to a phase by the geodesic equation

$$\frac{d^2 x^\mu}{d\tau^2} + \Gamma^\mu_{\nu\rho} \frac{dx^\nu}{d\tau} \frac{dx^\rho}{d\tau} = 0 \quad (4)$$

with initial positions $\tilde{x}_{\text{S}}^{(1)} \equiv x_{\text{S}} - x_{\text{M}}^{(1)}$ and $\tilde{x}_{\text{S}}^{(2)} \equiv x_{\text{S}} - x_{\text{M}}^{(2)}$. Writing down the solutions for the spatial coordinates in terms of the time coordinate $t = x^0$, we denote them by $\tilde{x}_{\text{S}}^{(1)}(t)$ and $\tilde{x}_{\text{S}}^{(2)}(t)$ respectively. The phase accumulated along the semi-classical path from spacetime point $A^{(i)} \equiv (0, \tilde{x}_{\text{S}}^{(i)})$ to $B^{(i)} \equiv (t, \tilde{x}_{\text{S}}^{(i)}(t))$ is given by[58]

$$\Phi^{(i)} = \int_{A^{(i)}}^{B^{(i)}} m_{\text{S}} d\tau = \int_{A^{(i)}}^{B^{(i)}} m_{\text{S}} \sqrt{-g_{\mu\nu} dx^\mu dx^\nu}, \quad (5)$$

where $m_{\text{S}}$ is the mass of S. Note that we use the metric convention $(-, +, +, +)$, in contrast to Stodolsky[58]. The total state of the

wavefunction after time $t$ is thus

$$|\psi(t)\rangle_{\text{MRS}}^{(\text{M})} = |\mathbf{0}\rangle_{\text{M}} \frac{1}{\sqrt{2}} \left( e^{-\frac{i}{\hbar}\Phi^{(1)}} \left|-\mathbf{x}_{\text{M}}^{(1)}\right\rangle_{\text{R}} \left|\tilde{\mathbf{x}}_{\text{S}}^{(1)}(t)\right\rangle_{\text{S}} + e^{-\frac{i}{\hbar}\Phi^{(2)}} \left|-\mathbf{x}_{\text{M}}^{(2)}\right\rangle_{\text{R}} \left|\tilde{\mathbf{x}}_{\text{S}}^{(2)}(t)\right\rangle_{\text{S}} \right). \quad (6)$$

To give a concrete example, consider the case of a spherically symmetric mass in the Newtonian limit, characterized by a weak gravitational field and non-relativistic particles such that we only need to take into account the time-time-component of the metric, $g_{00}(\mathbf{x}) = -1 - 2V(\mathbf{x}) = -1 + 2GM/|\mathbf{x} - \mathbf{x}_{\text{M}}|$. Given a probe particle with zero initial velocity and its initial positions in superposition forming one line with the location of the mass, the geodesics are

$$\tilde{x}_{\text{S}}^{(i)}(t) = \left( (\tilde{x}_{\text{S}}^{(i)})^{\frac{3}{2}} - 3\sqrt{\frac{MG}{2}} t \right)^{\frac{2}{3}} \quad (7)$$

along this line. The corresponding phase is obtained by integrating the line element over the semi-classical path. In the weak-field limit, we have $g_{\mu\nu} = \eta_{\mu\nu} + h_{\mu\nu}$ with $h_{00} = -2V(\mathbf{x})$ the only relevant component in the case of a Newtonian gravitational field. Following Stodolsky[58], the phase can be split into a special-relativistic contribution

$$\Phi_0^{(i)} = m_{\text{S}} \int_{A^{(i)}}^{B^{(i)}} \sqrt{-\eta_{\mu\nu} dx^\mu dx^\nu} = m_{\text{S}} \int_0^t \sqrt{1 - (\dot{\tilde{x}}_{\text{S}}^{(i)}(\tilde{t}))^2} \, d\tilde{t} \quad (8)$$

and a gravitational contribution

$$\varphi^{(i)} = m_{\text{S}} \int_0^t V\left(\tilde{x}_{\text{S}}^{(i)}(\tilde{t})\right) d\tilde{t} = -m_{\text{S}}\sqrt{2MG} \left( \sqrt[3]{(\tilde{x}_{\text{S}}^{(i)})^{3/2} - 3(\sqrt{MG/2})t} - \sqrt{\tilde{x}_{\text{S}}^{(i)}} \right). \quad (9)$$

While this concrete example pertains to the regime of semi-classical trajectories and weak gravitational fields, which is the relevant regime for table-top experiments, more general solutions could be considered equally well within this framework.

*Reference frame of R after time evolution.* Finally, by the principle of "covariance of dynamical laws under quantum coordinate transformations", applying the inverse QRF transformation $\hat{U}_T^\dagger = e^{-\frac{i}{\hbar}\hat{x}_{\text{M}}\hat{p}_{\text{S}}} \hat{\mathcal{P}}_{\text{RM}}$ yields the time-evolved state of M and S from the point of view of R,

$$|\psi(t)\rangle_{\text{RMS}}^{(\text{R})} = |\mathbf{0}\rangle_{\text{R}} \frac{1}{\sqrt{2}} \left( e^{-\frac{i}{\hbar}\Phi^{(1)}} \left|\mathbf{x}_{\text{M}}^{(1)}\right\rangle_{\text{M}} \left|\tilde{\mathbf{x}}_{\text{S}}^{(1)}(t) + \mathbf{x}_{\text{M}}^{(1)}\right\rangle_{\text{S}} + e^{-\frac{i}{\hbar}\Phi^{(2)}} \left|\mathbf{x}_{\text{M}}^{(2)}\right\rangle_{\text{M}} \left|\tilde{\mathbf{x}}_{\text{S}}^{(2)}(t) + \mathbf{x}_{\text{M}}^{(2)}\right\rangle_{\text{S}} \right). \quad (10)$$

In this frame of reference, the particle gets entangled with the massive body, moving in a superposition of trajectories in the potential sourced by a mass at $\mathbf{x}_{\text{M}}^{(1)}$ and $\mathbf{x}_{\text{M}}^{(2)}$ respectively.

This result is consistent with the assumption that the solution comes from a "linear combination" of gravitational fields, as discussed in Christodoulou and Rovelli's analysis of the Bose-Marletto-Vedral proposal[18]. Assuming that we can associate a semi-classical state $|g\rangle$ to the gravitational field sourced by the mass M in each branch and that the time evolution of other systems is governed by the field in each branch separately while taking into account the relative phase caused by the different gravitational fields, one arrives at the same conclusion. We, however, derive this result from the principle of "covariance of dynamical laws under quantum coordinate transformations", rather than making assumptions about the Hilbert space structure of gravitational field states or the dynamics in the presence of superpositions thereof. In other words, we argue that the question of whether the gravitational field is quantized or not must be answered affirmatively if Einstein's equations satisfy the extended symmetry principle.

We would also like to stress again that the fact that the two configurations can be related by a simple spatial shift does not imply that the gravitational field was the same in each branch from the start. As already pointed out in the introduction, such an argument disregards the presence of the probe particle S and the reference system R. It breaks the equivalence by giving physical meaning to the spacetime points of their locations and to the proper distances between them and the gravitational source. These proper distances are diffeomorphism invariant quantities, which are in a true superposition in the example considered in this section. If the massive object was floating in empty space, its positions and hence superpositions thereof would be meaningless. However, as soon as other objects are introduced, the position of the mass regains physical significance relative to these other objects and we can meaningfully speak of superpositions[53].

**Generalization to N masses**. In the following, we generalize the above argument to the case of $N$ gravitating point masses in a superposition of configurations. In this section, we are going to work within the weak-field (i.e. Newtonian) limit only. A more general approach can be found in the "Methods" and "Relative-Distance-Preserving Transformation Using an Auxiliary System" sections. Note that we have to restrict to the case in which these configurations are related by relative-distance-preserving transformations. In the weak-field limit, this means that the mass configurations are related via translations and rotations, i.e. Euclidean isometries. This restriction is required since we do not want to work with the metric directly, as discussed in the introduction. If we were to, this would require the formulation of a rigorous mathematical framework, including a precise notion of a Hilbert space structure for the metric together with a well-defined inner product. Furthermore, this would potentially require the extension of QRFs to quantum fields in order to change the metric locally in each spacetime point. However, the aim of this work is not to build such a framework but to provide an argument which allows to circumvent the issue of handling superpositions of metric states altogether. This can be achieved by restricting relative-distance-preserving transformations.

Consider the system M consisting of $N$ gravitating point masses, a quantum system S in an arbitrary state and a reference frame R. It is crucial that R carries enough degrees of freedom in order to uniquely specify the transformation that maps the state relative to R to the one relative to M. In the weak-field limit, this amounts to R being made up of two particles (see Fig. 3). The constituents of R will serve as an indicator of direction, specifying the origin and one axis of the reference frame R. Let us start with the following state:

$$\left|\psi(t)\right\rangle_{\text{RMS}}^{(\text{R})} = \left|\mathbf{0}\right\rangle_{\text{R}_1}\left|\mathbf{e}_1\right\rangle_{\text{R}_2} \otimes \frac{1}{\sqrt{K}}\left(\sum_{i=1}^{K}\left|\mathbf{x}_1^{(i)}\right\rangle\left|\mathbf{x}_2^{(i)}\right\rangle\cdots\left|\mathbf{x}_N^{(i)}\right\rangle\right)_{\text{M}} \otimes \left|\phi(t)\right\rangle_{\text{S}},$$
(11)

where $\text{R}_i, i = 1, 2$ denote the different subsystems of R and $\mathbf{x}_j^{(i)}$ is a three-vector denoting the position of the subsystem $\text{M}_j$ in Euclidean space. The subscripts $1, \ldots, N$ indicate the subsystem of M under consideration while the superscripts $(i), i = 1, \ldots, K$ denote the branch of the superposition. Also, $\mathbf{e}_1$ indicates the main axis marked by the vector from $\text{R}_1$ to $\text{R}_2$ (see Fig. 3). The time $t$ denotes the coordinate time; in particular, it can operationally be seen as the proper time of the reference system R. The conditions for this coordinate time to be well-defined as the proper time of R are discussed in Supplementary Note 2.

Again, the idea is to perform a quantum change of coordinates, such that in the new coordinate system with respect to system M, the configuration of M is definite. Here, to provide more intuition for the motion of the probe, we make use of the semi-classical

approximation to treat the trajectory of the probe. Keep in mind however that our method goes beyond this approximation. Starting from the description in the new coordinate system, we can use the geodesic equation together with the phase in Eq. (5) to compute the semi-classical trajectories of a freely falling probe S, as in the previous section. Since we are restricting to the weak-field limit in this section, however, we can also make use of the explicit Hamiltonian to determine the general motion for an arbitrary quantum state of S. After time evolution, we apply the inverse QRF transformation to transform back into the original frame.

Let us quickly outline the strategy to make the mass configuration definite. Note, however, that the detailed calculations are given in "Methods" section, in the subsection "Explicit calculation for the general case of $N$ masses". The transformation consists of four steps: First, we change into more easily tractable relative coordinates by applying $\hat{T}_{\text{rel}}^{(\text{M})}$. In particular, this maps the states of the first few subsystems of M to relative coordinates $\mathbf{a} = \mathbf{x}_2 - \mathbf{x}_1, \mathbf{b} = \mathbf{x}_3 - \mathbf{x}_1, \mathbf{c} = \mathbf{x}_4 - \mathbf{x}_1$. Secondly, we perform a controlled shift and rotation $\hat{U}_{\text{MSR}}$ on all systems M, S, and R. Then, we apply a generalized parity swap $\hat{\mathcal{P}}_{\text{MR}}$ which exchanges the labels of $\text{M}_1$ and $\text{M}_2$ with $\text{R}_1$ and $\text{R}_2$ respectively, implements reflections and thus assigns M the role of the reference frame. Finally, $\hat{T}_{\text{rel}}^{(\text{M})\dagger}$ transforms back to the original type of coordinates, but now relative to the new frame. The full QRF change operator is given by

$$\left(\hat{\mathcal{S}}^{\text{R}\rightarrow\text{M}}\right)^{\dagger} = \hat{T}_{\text{rel}}^{(\text{M})\dagger} \circ \hat{\mathcal{P}}_{\text{MR}} \circ \hat{U}_{\text{MSR}} \circ \hat{T}_{\text{rel}}^{(\text{M})}.$$
(12)

Note that this operator is unitary since it is a composition of unitary operators and consequently preserves the observed probabilities. A concrete example to illustrate the action of the QRF change operator $\hat{\mathcal{S}}^{\text{R}\rightarrow\text{M}}$ can be found in Supplementary Note 3. Applying the operator to the initial state in Eq. (11) gives rise to the following state:

$$\left|\psi(t)\right\rangle_{\text{MRS}}^{(\text{M})} = \hat{\mathcal{S}}^{\text{R}\rightarrow\text{M}}\left|\psi\right\rangle_{\text{RMS}}^{(\text{R})} = \left|\mathbf{0}\right\rangle_{\text{M}_1}\left|\mathbf{f}_1\right\rangle_{\text{M}_2}\left|\widetilde{\mathbf{b}}\right\rangle_{\text{M}_3}\left|\widetilde{\mathbf{c}}\right\rangle_{\text{M}_4}\cdots\left|\widetilde{\mathbf{x}}_N\right\rangle_{\text{M}_N}$$

$$\otimes \frac{1}{\sqrt{K}}\sum_{i=1}^{K}\left(\left|-\mathbf{x}_1^{(i)}\right\rangle_{\text{R}_1}\left|F_{\mathbf{e}_1}\mathbf{a}^{(i)}\right\rangle_{\text{R}_2} \otimes \left|\widetilde{\phi}^{(i)}\right\rangle_{\text{S}}\right),$$
(13)

where $\mathbf{f}_1$ denotes a length-adjusted version of $\mathbf{e}_1$, the vectors $\widetilde{\mathbf{b}}, \widetilde{\mathbf{c}}, \widetilde{\mathbf{x}}_i$, and $\widetilde{\phi}^{(i)}$ are obtained from $\mathbf{b}, \mathbf{c}, \mathbf{x}_i$, and $\phi^{(i)}$ through rotations and shifts, and $F_{\mathbf{e}_1}\mathbf{a}^{(i)}$ corresponds to $\mathbf{a}^{(i)}$ flipped across the $\mathbf{e}_1$-axis. We see that, as expected, relative to M, system M is itself in a definite configuration and the state of the quantum system S becomes entangled with system R. Now, it is possible to write down the Hamiltonian determining the motion of system S. We again assume that the masses in this set-up are static and do not evolve dynamically with respect to R. As a consequence, to obtain the time evolution of the state of all systems relative to the reference frame associated with system M, it is enough to compute the time-evolved state of system S,

$$\left|\widetilde{\phi}^{(i)}(t+\Delta t)\right\rangle_{\text{S}} = e^{-\frac{i}{\hbar}\hat{H}_{\text{SR}}^{(\text{M})}\Delta t}\left|\widetilde{\phi}^{(i)}(t)\right\rangle_{\text{S}},$$
(14)

using the Hamiltonian

$$\hat{H}_{\text{SR}}^{(\text{M})} = \frac{\hat{\boldsymbol{\pi}}_{\text{S}}^2}{2m_{\text{S}}} + m_{\text{S}}\hat{V}(\hat{\mathbf{q}}_{\text{S}}) = \frac{\hat{\boldsymbol{\pi}}_{\text{S}}^2}{2m_{\text{S}}} - m_{\text{S}}\sum_{i=1}^{N}\frac{GM_i}{|\hat{\mathbf{q}}_{\text{S}} - \mathbf{q}_{\text{M}_i}|}$$
(15)

in the Newtonian gravity limit. Finally, using the principle of "covariance of dynamical laws under quantum coordinate transformations", we can apply the inverse QRF transformation $\left(\hat{\mathcal{S}}^{\text{R}\rightarrow\text{M}}\right)^{\dagger} = \hat{\mathcal{S}}^{\text{M}\rightarrow\text{R}}$ which gives the time-evolved state relative to

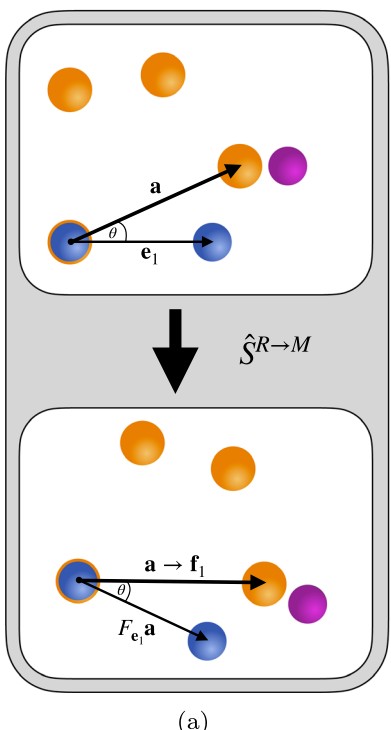

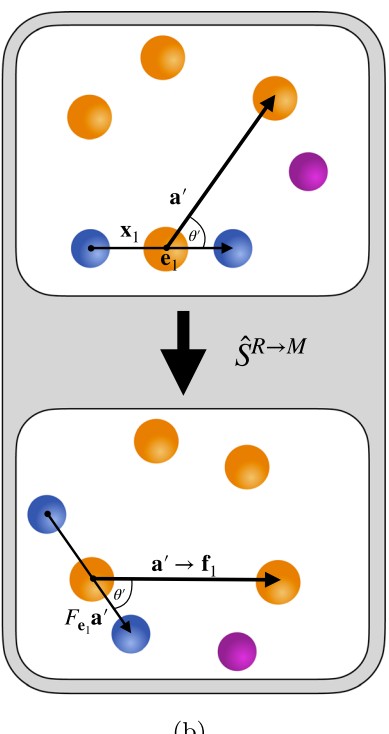

(a)  (b)

**Fig. 3 Superposition of two mass configurations.** We consider four massive objects M (in orange), the reference system R (in blue) and a system S (in violet). The reference R consists of two particles; this is because it serves as a reference frame for both position and orientation in space. The subfigures (**a**) and (**b**) depict the configuration in two different branches and the quantum reference frame transformations therein. The upper configurations are given in the reference frame of R while the lower ones are given in the frame of M. To go from one frame to the other, the quantum reference frame transformation $\hat{S}^{R \to M}$ is applied. This transformation controls on the angle $\theta$ between the axes $\mathbf{e}_1$ and $\mathbf{a}$ and rotates the entire configuration by $-\theta$. Furthermore, it controls on $\mathbf{x}_1$ and shifts the entire configuration by $-\mathbf{x}_1$ such that one of the massive objects is in the origin. The end result is that, although the masses are in a superposition of positions in the frame of R (top right and top left), all masses are in a definite position for all amplitudes in the frame of M (bottom depiction in **a** and bottom depiction in **b**.

the original frame of system R:

$$\left| \psi(t+\Delta t) \right\rangle_{\mathrm{RMS}}^{(R)} = |\mathbf{0}\rangle_{R_1} |\mathbf{e}_1\rangle_{R_2} \otimes \frac{1}{\sqrt{K}} \left( \sum_{i=1}^{K} \left| \mathbf{x}_1^{(i)} \right\rangle_{M_1} \left| \mathbf{x}_2^{(i)} \right\rangle_{M_2} \cdots \left| \mathbf{x}_N^{(i)} \right\rangle_{M_N} \otimes \left| \phi^{(i)}(t+\Delta t) \right\rangle_{S} \right).$$
(16)

As one can see from this, the state of system S becomes entangled over time with the mass configuration M.

Note that it is possible to go beyond the Newtonian case, i.e. beyond Euclidean isometry transformations. For this, one requires a more general operator that can implement general coordinate transformations. For the sake of the argument we make in this article, these transformations should still be relative-distance-preserving transformations. The precise form of such a more general operator is given in the "Methods" section "Relative-distance-preserving transformation using an auxiliary system" section. Note that the specific operator that we find requires adding an ancilla system whose quantum state marks the distinct branches of the superposition. At this current stage, it is only suitable for situations that are physically different from those considered so far, namely for set-ups which involve an additional system that can serve as the ancilla. The existence of such an auxiliary system is compatible with some experimental proposals, e.g. those utilizing a Stern-Gerlach set-up to create massive superpositions[23]. However, most other proposals do not include such an additional system and therefore no such degree of freedom. In any case, the experimental set-ups realizable in the near future rely on the weak-field limit and thus the operator given in the present section is suitable.

**Application: time dilation.** We can apply the same argument to study the behavior of a clock in the presence of a massive object in superposition. To this end, consider a single mass M in a superposition of two spatial locations, a clock C and a remote initial reference system R, whose proper time can be identified with the coordinate time $t$. A complementary situation was considered in Zych et al.[59,60], where a clock moving in a superposition of trajectories in a definite metric generated by a large mass in the laboratory frame was studied. The clock was found to exhibit a superposition of time dilations as its trajectories passed through different gravitational potentials. To test an extension of the Einstein equivalence principle to QRFs, Giacomini[45] and Cepollaro and Giacomini[61] considered a similar situation by moving to the reference frame of the clock, where the laboratory and thus the mass are in a spatial superposition. Here we consider a related yet different situation, where the clock is localized and the large mass is in a superposition in the reference frame of the laboratory, so that the metric is indefinite. Following Zych et al.[62], we take the clock to be a two-level system with an external degree of freedom specifying its position and an internal degree of freedom describing its energy, which can take values $E_0$ or $E_1$ associated to eigenstates $|0\rangle$ and $|1\rangle$ respectively. Hence, the internal Hamiltonian is given by $\hat{\Omega} = E_0 |0\rangle\langle 0| + E_1 |1\rangle\langle 1|$, although more general Hamiltonians could be considered as well. Note that as an internal property, the superposition state of energies evolves with the proper time of the clock in its rest frame – just as the decay rate of a relativistic particle is given in terms of the proper time with respect to its rest frame. Finally, we assume that the positions of the mass and the clock are held fixed with

respect to the reference system throughout the experiment, for example through a strong potential, and focus on the evolution of the clock's internal energy.

Consider two events, $\mathcal{E}_0$ and $\mathcal{E}_1$, such as reading the position of the hands of a clock, and denote by $t_0$ the coordinate time assigned to $\mathcal{E}_0$. Due to the functional dependence of the proper time $\tau$ of the clock on the coordinate time $t$, we can further associate a proper time $\tau(t_0) = \tau_0$ to this event. At this time, the internal state of the clock is initialized to $|s(\tau_0)\rangle = \frac{1}{\sqrt{2}}(|0\rangle + |1\rangle)$. The initial state of the composite system is then given by

$$|\Psi(t_0)\rangle_{\mathrm{RMC}}^{(\mathrm{R})} = |\mathbf{0}\rangle_{\mathrm{R}} \frac{1}{\sqrt{2}} \left( \left|\mathbf{x}_{\mathrm{M}}^{(1)}\right\rangle_{\mathrm{M}} + \left|\mathbf{x}_{\mathrm{M}}^{(2)}\right\rangle_{\mathrm{M}} \right) |\mathbf{x}_{\mathrm{C}}\rangle_{\mathrm{C}_{\mathrm{ext}}} |s(\tau_0)\rangle_{\mathrm{C}_{\mathrm{int}}}. \tag{17}$$

The second event $\mathcal{E}_1$ marks the reading of the clock's internal state and occurs at a later coordinate time $t$. To determine the state $|\psi(t)\rangle$, we need to find the proper time $\tau$ elapsed between coordinate time $t_0$ and $t$. This, however, depends on the gravitational field at the position of the clock, which is not well-defined in the reference frame of R. We thus change into the reference frame of M using the same transformation as in "The case of one massive object" section and obtain

$$\begin{aligned}|\psi(t_0)\rangle_{\mathrm{MRC}}^{(\mathrm{M})} = |\mathbf{0}\rangle_{\mathrm{M}} \frac{1}{\sqrt{2}} &\left( \left|-\mathbf{x}_{\mathrm{M}}^{(1)}\right\rangle_{\mathrm{R}} \left|\mathbf{x}_{\mathrm{C}} - \mathbf{x}_{\mathrm{M}}^{(1)}\right\rangle_{\mathrm{C}_{ext}} \right. \\ &\left. + \left|-\mathbf{x}_{\mathrm{M}}^{(2)}\right\rangle_{\mathrm{R}} \left|\mathbf{x}_{\mathrm{C}} - \mathbf{x}_{\mathrm{M}}^{(2)}\right\rangle_{\mathrm{C}_{ext}} \right) |s(\tau_0)\rangle_{\mathrm{C}_{\mathrm{int}}}. \end{aligned} \tag{18}$$

In this frame, the mass is in a definite position while the clock is in a superposition of two positions, $\tilde{\mathbf{x}}_{\mathrm{C}}^{(1)} \equiv \mathbf{x}_{\mathrm{C}} - \mathbf{x}_{\mathrm{M}}^{(1)}$ and $\tilde{\mathbf{x}}_{\mathrm{C}}^{(2)} \equiv \mathbf{x}_{\mathrm{C}} - \mathbf{x}_{\mathrm{M}}^{(2)}$. This is precisely the situation discussed in Zych et al.[62]. Since the gravitational field is definite in this frame, one can determine the elapsed proper time $\tau^{(i)}$ in each branch $i = 1, 2$. In the Newtonian limit, it is

$$\tau^{(i)}(\tilde{\mathbf{x}}_{\mathrm{C}}^{(i)}, t) = t \left( 1 + \frac{V(\tilde{\mathbf{x}}_{\mathrm{C}}^{(i)})}{c^2} \right). \tag{19}$$

To obtain the time-evolved state, we further have to take into account the dynamics of the external degrees of freedom. While we assume that they are kept fixed throughout the duration of the experiment, the state nevertheless accumulates a phase $\Phi^{(i)}$ determined by Eq. (5) for constant $\tilde{\mathbf{x}}^{(i)}$ in each branch. Note that the phase $\Phi^{(i)}$ can be split into a gravitational part $\varphi^{(i)}$ and a special relativistic contribution, the former of which corresponds to the COW (Colella-Overhauser-Werner) phase $\varphi^{(i)} = m_{\mathrm{S}} g |x^{(i)}| t$[55]. Since the special relativistic phase depends only on the velocity of the probe, which is kept zero in our set-up, it is the same in both branches and can thus be omitted. The time-evolved state is therefore

$$\begin{aligned}|\psi(t)\rangle_{\mathrm{MRC}}^{(\mathrm{M})} = |\mathbf{0}\rangle_{\mathrm{M}} \frac{1}{\sqrt{2}} &\left( \left|-\mathbf{x}_{\mathrm{M}}^{(1)}\right\rangle_{\mathrm{R}} e^{-\frac{i}{\hbar}\Phi^{(1)}} \left|\tilde{\mathbf{x}}_{\mathrm{C}}^{(1)}\right\rangle_{\mathrm{C}_{ext}} |s(\tau_0 + \tau^{(1)})\rangle_{\mathrm{C}_{\mathrm{int}}} \right. \\ &\left. + \left|-\mathbf{x}_{\mathrm{M}}^{(2)}\right\rangle_{\mathrm{R}} e^{-\frac{i}{\hbar}\Phi^{(2)}} \left|\tilde{\mathbf{x}}_{\mathrm{C}}^{(2)}\right\rangle_{\mathrm{C}_{ext}} |s(\tau_0 + \tau^{(2)})\rangle_{\mathrm{C}_{\mathrm{int}}} \right), \end{aligned} \tag{20}$$

where $|s(\tau_0 + \tau^{(i)})\rangle = e^{-i\hat{\Omega}\tau^{(i)}} |s(\tau_0)\rangle$. Finally, we change back to the original reference frame with the inverse QRF transformation to obtain

$$|\psi(t)\rangle_{\mathrm{RMC}}^{(\mathrm{R})} = |\mathbf{0}\rangle_{\mathrm{R}} \frac{1}{\sqrt{2}} \left( e^{-\frac{i}{\hbar}\Phi^{(1)}} \left|\mathbf{x}_{\mathrm{M}}^{(1)}\right\rangle |s(\tau_0 + \tau^{(1)})\rangle_{\mathrm{C}_{\mathrm{int}}} + e^{-\frac{i}{\hbar}\Phi^{(2)}} \left|\mathbf{x}_{\mathrm{M}}^{(2)}\right\rangle |s(\tau_0 + \tau^{(2)})\rangle_{\mathrm{C}_{\mathrm{int}}} \right) |\mathbf{x}_{\mathrm{C}}\rangle_{\mathrm{C}_{\mathrm{ext}}}. \tag{21}$$

Note that the external state of the clock factorizes out because we trap the clock to stay at the same spatial location. The internal

degree of freedom, on the other hand, gets entangled with the mass due to different time dilations in each branch, which in turn derive from the difference in relative distance between the clock and the mass. Let us stress that in the situation described here, we have a genuine superposition of an observable quantity, independent of the reference frame: the proper time of the clock (Fig. 4). Moreover, the time dilation at the heart of this superposition is a universal effect, depending only on the spacetime and the position of the clock in it and not on the internal structure of the clock or the nature of the non-gravitational force that controls its ticking. In this sense, the superposition of gravitational time dilation of a clock located on a single spacetime trajectory (in the reference frame of R) is a genuine phenomenon due to a non-classical spacetime. While it is possible to reproduce this effect in a definite spacetime if the clock is set to follow a superposition of two different trajectories, this is impossible for a clock following a single trajectory as we consider here.

As a rough estimate for the difference in time dilation, consider M to be a solid state sphere of $10^{-8}$kg. Taking the distance between M and S to be $l^{(1)} = 5.0 \times 10^{-5}$m and $l^{(2)} = 5.5 \times 10^{-5}$m in the first and second branch respectively and considering a time evolution for $t = 1$s, we find that $\Delta\tau = \tau^{(2)} - \tau^{(1)} \approx 10^{-32}$ s. Although this time is extremely small compared to the best atomic clocks of today, it is many orders of magnitude larger than the Planck time ($5.31 \times 10^{-44}$ s). This shows that effects due to spacetime superpositions can occur well before the typical Planck scale[63].

Of course, any experiment that measures the gravitational effects of massive superpositions will face significant challenges. Firstly, one would need to suppress the gravitational and non-gravitational fields sourced by any object other than the gravitational source under consideration. By carefully controlling the gravitational field of the environment, such that there are no significant differences across the branches of the superposition, one should be able to isolate the effects sourced by the massive configuration. Furthermore, due to the interaction with an external environment, the gravitational source and the probe particle can become entangled with their surroundings and may lose their quantum coherence. Many specific models in which a system interacts with its environment have been studied including collisional and thermal decoherence[64–66] as well as the decoherence induced by the gravitational field background[67–70]. Any experiment will have to be performed within a time frame shorter than the decoherence time.

## Discussion

In this work, we provide a rigorous argument that allows us to make predictions for situations in which the gravitational source is not in a classical configuration but in a quantum superposition thereof. Using a quantum reference frame transformation, we can change into the frame associated with the massive object, in which the gravitational field is definite, and use known physics to solve concrete problems and in particular determine the time evolution of objects in the presence of the mass configuration. Assuming the "covariance of dynamical laws under quantum coordinate transformations", applying the inverse transformation to the evolved state yields the dynamical description in the original frame. This procedure does not a priori rely on a theory for the gravitational field sourced by an object in a quantum superposition or on assigning orthogonal quantum states to the gravitational field. We make the important observation that this requires the gravitational source to be in a superposition of configurations related by relative-distance-preserving transformations. If this is not the case, the QRF transformation does not

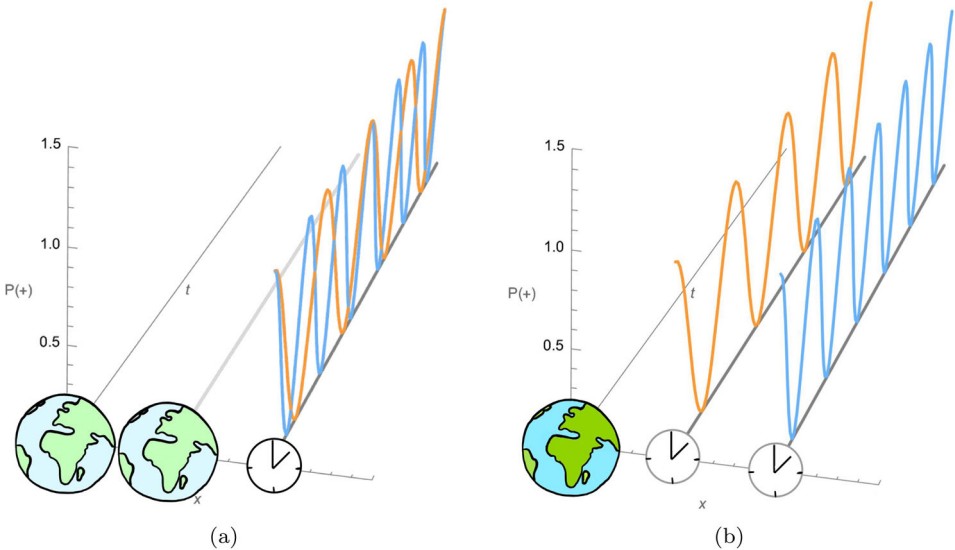

**Fig. 4 Qualitative depiction of the behavior of a quantum clock.** Subfigure (**a**) depicts the description relative to the reference frame of R (not depicted explicitly). Subfigure (**b**) depicts the description in the reference frame of the gravitational source M. In the latter frame, the massive object is in a definite state while the clock is in a superposition of spatial locations (x-axis). It ticks – that is, oscillates between the states $|-\rangle$ and $|+\rangle$ – at different rates. The figure shows the probability to find it in the $|+\rangle$-state over time. The latter is given by $P_+^{(i)}(t) = \cos^2\left(\frac{E_0-E_1}{2}\left(1 - \frac{GM}{c^2}\frac{1}{x_M^{(i)}}\right)t\right)$ and depends on the energy levels $E_0$ and $E_1$ of the clock as well as the mass $M$ and the position $x_M^{(i)}$ of the gravitational source. This superposition of ticking rates carries over to the reference frame of R, in which the clock is in a definite position while the gravitational source is in a spatial superposition.

constitute a symmetry of the physical situation. This is analogous to – and in fact a generalization of – the familiar situation in Newtonian physics, in which changing into a rotating or accelerating frame of reference will not leave covariant the dynamical laws and in particular the form of the gravitational force.

For the case of Newtonian gravity, we provide an explicit operator in "The case of one massive object" and "Generalization to N masses" sections which transforms states in the reference frame of R to states in the reference frame of the system of massive objects M. We discuss set-ups in the semi-classical approximation and beyond. In particular, we determine the motion of a probe particle following semi-classical trajectories and further employ the Hamiltonian formalism to compute the evolution of any quantum mechanical probe system in a general state in the "Transformation of the Hamiltonian operator" section. In the "Application: time dilation" section we give an application of the above argument and transformations: we consider a simple quantum clock in the presence of one mass in a superposition of different spatial locations and show, using QRF transformations and time evolution in the frame with a definite spacetime, that this gives rise to a superposition of time dilations of the clock. While the resulting difference in proper times is still outside the reach of current atomic clocks, it is many orders of magnitude larger than the Planck time, showing that effects due to spacetime superpositions can occur well before the typical Planck scale, at which quantum gravity effects are usually expected to manifest. In "Relative-distance-preserving transformation using an auxiliary system" we present a more general QRF change operator that can be applied to configurations with general spacetime metrics and thus extends beyond the weak-field/perturbative regime. Note that this transformation makes use of the semi-classical approximation for the probe S, i.e. the probe moves in a superposition of semi-classical trajectories.

The main argument provided in this paper is of a very general nature – it extends our ability to solve any problem in presence of a classical gravitational source to situations in which the gravitational source is in a superposition of configurations related by

relative-distance-preserving transformations. We thus expect that it can equally well be applied to situations in which the probe is described by a quantum field. While the current framework of QRF transformations does not extend to quantum fields at this moment, the principle of "covariance of dynamical laws under quantum coordinate transformations" can still be applied in this context. The construction of a transformation operator that can be applied to quantum fields thus provides a fruitful direction for future work. Unlike many other works on the QRF formalism, we do not only construct a framework but also provide explicit tools to solve concrete problems. As a first step, we apply them to predict the dynamics of probe particles in the presence of gravitational sources in superposition, but we are confident that their application goes beyond the scenarios considered here.

Besides its application to concrete physical problems, our approach also provides a posteriori a justification for developing a quantum formalism for the gravitational field. One of the long-standing problems in fundamental physics is whether the gravitational field is quantized or not. In the absence of experimental evidence, theoretical arguments have been advanced for both possibilities. Our results can be interpreted to mean that, assuming that Einstein's equations satisfy the extended symmetry principle, quantization of gravity is necessary in the sense that general relativity holds in each branch of the superposition separately. In particular, one could assign a state $|x_M, g\rangle$ to describe the position of M in each branch and the gravitational field degrees of freedom enslaved by the spatial location of the massive object(s)[71]. That is, we consider only the degrees of freedom determined entirely by the source mass, excluding any independent degrees of freedom associated with gravitational waves, and consequently do not include any gravitons in the quantum description. Specifically, the metric $g$ in $|x_M^{(i)}, g\rangle$ is the classical solution sourced by a mass stationary in the position $x_M^{(i)}$ in the $i$-th branch. These states $|x_M^{(i)}, g\rangle$ can be linearly superposed and are orthogonal for different $x^{(i)}$, given the classical

distinguishability of the mass distributions. Thus, in future work, the argument provided in this work can be used to justify the assignment of orthogonal quantum states to classically distinguishable gravitational fields[43,49] – keeping in mind the condition that the superposed mass configurations are related by relative-distance-preserving transformations. It also vindicates a short-cut for determining the dynamics in the presence of massive objects in superposition by assuming a superposition of gravitational fields in the above sense from the start.

The superposition of gravitational fields is also in line with the ansatz of linearized quantum gravity[5], in which perturbations of the metric are quantized. However, we go beyond the regime covered by this theory as we are not restricted to the perturbative regime. In particular, we can consider superpositions of gravitational fields (in the above sense) which are significantly different at a given spacetime point, as opposed to being related by a perturbative contribution (Supplementary Note 4). Moreover, under the restricted set of mass configurations studied here, our approach allows us to transform to a frame in which the spacetime becomes globally definite, contrary to recent constructions of a local quantum inertial frame in which the metric becomes definite only in the origin of the reference frame[43,49]. This feature of our transformation allows us to describe the entire spacetime trajectories of probe particles within spacetimes in superposition, which is one of the paradigmatic problems of quantum gravity and the one most likely to be first implemented experimentally in the future.

Furthermore, we start from a more fundamental assumption, namely "covariance of dynamical laws under quantum coordinate transformations", instead of making any a priori statement about the quantum nature of the gravitational field. If future experimental results can confirm predictions based on our argument, this serves as corroboration for the underlying extended symmetry principle. Moreover, if the predictions made by other proposals, based on the assumption of a linearly superposed gravitational field in the above sense, are confirmed experimentally, this can be seen as indirect evidence for the principle of covariance under quantum coordinate transformations as well. On the other hand, our predictions stand in stark contrast with proposals such as semi-classical quantum gravity and gravitational collapse models[9–11]. The former predicts a definite spacetime sourced by a gravitational source in superposition and thus avoids any entanglement between the probe and the massive objects as well as any superposition in the path of the particle. The latter predicts that a superposition of gravitational fields must necessarily collapse and thus entanglement and superposition cannot be sustained. As illustrated in Fig. 5, this is in contradiction with the principle of "covariance of dynamical laws under quantum coordinate transformations". There is an inherent asymmetry between a situation in which a light probe particle is localized in the presence of a large massive object in superposition and one in which the probe is in a superposition while the large mass is localized. Given that this is the same situation described in different frames, as argued in this paper, we see that these proposals must implicitly assume a preferred reference frame[24].

Concerning future research directions, our work can be extended in different ways. The most pressing issue concerns a rigorous construction of a QRF framework for quantum fields. Based on this, one could study many more interesting effects within the realm of quantum field theory on curved spacetime, e.g. the Unruh effect in a superposition of spacetimes. Secondly, there are several assumptions made in this article that could be lifted in future work. This includes letting the reference frame R interact gravitationally with the source masses and, possibly, considering a backreaction of the probe system on the metric.

With regards to the massive objects, an interesting extension of the current work would be to go beyond semi-classical states and including a non-trivial time evolution with respect to the reference system. Thirdly, we believe that there is much room to go beyond the applications discussed above and use the tools provided in this article to design experimental proposals within the reach of near-future table-top experiments, probing the quantum nature of spacetime. Given the rapid advances in quantum technologies, which open up new avenues for testing the gravitational properties of quantum matter, we believe that important developments on the theoretical side are crucial to understand the implications of such experiments for the notion of spacetime in the quantum regime. Our work is a step forward in this direction. Finally, in line with the construction of the general theory of relativity in which the covariance under general coordinate transformations played a crucial role[72], the "covariance of physical laws under quantum coordinate transformations" can serve in the long term as a guiding principle in the construction of a proper theory of quantum gravity.

## Methods

**Transformation of the Hamiltonian operator.** Consider first a single mass in superposition of $K$ different positions. If we assume that the gravitational field sourced by the mass is weak and that the particle is not moving with relativistic velocities, we can work within the Newtonian approximation in which the only deviation from Minkowski spacetime manifests in the time-time-component of the metric, $g_{00} = -1 - 2V(\mathbf{x})$ with $V(\mathbf{x}) = -GM/|\mathbf{x} - \mathbf{x}_M|$. In this case, the geodesic equation simplifies to Newton's equation,

$$m\frac{d^2\mathbf{x}}{dt^2} = -m\nabla V(\mathbf{x}). \tag{22}$$

In order to determine the unitary transformation implementing the time translation of the quantum state of the particle, we need to write down the Hamiltonian governing the temporal evolution. To avoid confusion between operators defined with respect to different reference frames, we denote by $\hat{\mathbf{x}}$ and $\hat{\mathbf{p}}$ the three-dimensional position and momentum operators with respect to R while using $\hat{\mathbf{q}}$ and $\hat{\boldsymbol{\pi}}$ in the reference frame of M. Assuming that the dynamics of the reference frame can be neglected, we obtain the Hamiltonian in the reference frame of M

$$\hat{H}_{RS}^{(M)} = \frac{\hat{\boldsymbol{\pi}}_S^2}{2m_S} + m_S\hat{V}(\hat{\mathbf{q}}_S) \tag{23}$$

where $\hat{V}(\hat{\mathbf{q}}_S)$ is the gravitational potential, which, for one point mass, takes the form $\hat{V}(\hat{\mathbf{q}}_S) = -\frac{GM}{|\hat{\mathbf{q}}_S - \mathbf{q}_M|}$. Note that since we are working in the frame of M, we have $\mathbf{q}_M = 0$. In the above equation, identity transformations on the subspaces of the reference system and the mass are left implicit. The unitary which transforms M's into R's frame of reference is

$$\hat{U}_T^\dagger = e^{-\frac{i}{\hbar}\hat{\mathbf{x}}_M\hat{\mathbf{p}}_S}\hat{\mathcal{P}}_{MR}^\dagger = \hat{\mathcal{P}}_{RM}e^{\frac{i}{\hbar}\hat{\mathbf{q}}_R\hat{\boldsymbol{\pi}}_S}. \tag{24}$$

Applying this to the Hamiltonian above, we obtain

$$\hat{H}_{MS}^{(R)} = \hat{U}_T^\dagger\hat{H}_{RS}^{(M)}\hat{U}_T = \hat{\mathcal{P}}_{RS}\frac{\hat{\boldsymbol{\pi}}_S^2}{2m_S}\hat{\mathcal{P}}_{RM}^\dagger + \hat{\mathcal{P}}_{RM}\left(\sum_{n=0}^\infty \frac{1}{n!}\left[\frac{i}{\hbar}\hat{\mathbf{q}}_R\hat{\boldsymbol{\pi}}_S, m_SV(\hat{\mathbf{q}}_S)\right]_n\right)\hat{\mathcal{P}}_{RM}^\dagger \tag{25}$$

where $[X, Y]_n = [X, [X, Y]_{n-1}]$ and $[X, Y]_0 = Y$. If the potential is infinitely differentiable, we can use

$$\left[\frac{i}{\hbar}\hat{\mathbf{q}}_R\hat{\boldsymbol{\pi}}_S, V(\hat{\mathbf{q}}_S)\right]_n = \hat{\mathbf{q}}_R^n V^{(n)}(\hat{\mathbf{q}}_S), \tag{26}$$

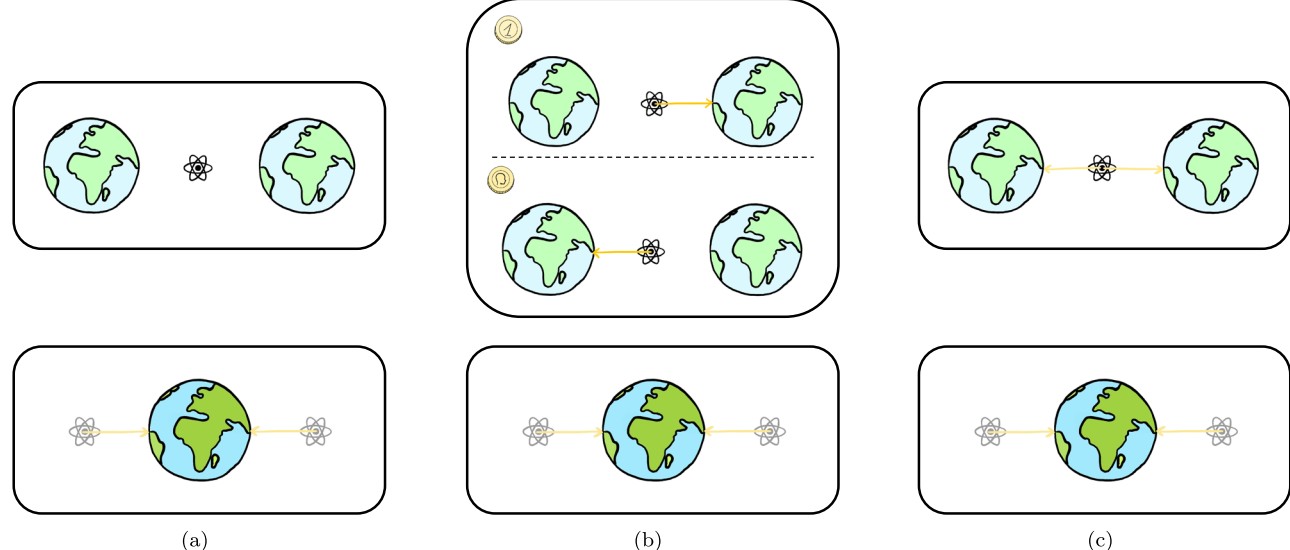

**Fig. 5 Comparison of predictions of different approaches to quantum gravity.** We consider a massive object in superposition of two locations with a probe located exactly in the middle of these locations in the reference frame of R (not explicitly depicted here). The upper subfigures illustrate the situation in the original frame while the lower subfigures show the situation in the reference frame of the massive object M. **a** Semi-classical gravity predicts an effective classical gravitational field, obtained from the expectation value of the matter degrees of freedom, resulting in a vanishing potential at the location of the probe. As a result, it remains stationary. **b** According to gravitational collapse models, the state of the massive object collapses into a definite position state after a short time. Conditioned on the outcome of the collapse (indicated by the face of the coin), the probe either moves to the left or the right. **c** Using the argument presented in this article, we find the probe in a superposition of moving to the left and to the right, entangled with the position of the mass. In the frame of M, all three approaches would make the same predictions: the probe, which starts in a superposition of positions, moves in a superposition of trajectories towards the mass. This is in agreement with the fact that all three theories are compatible with each other when light massive objects are spatially superposed. However, only the trajectories in in the third case are in accordance with the motion in the frame of M and thus respect the principle of "covariance of dynamical laws under quantum coordinate transformations''.

which can be proven straightforwardly via induction to obtain

$$\hat{H}_{\mathrm{MS}}^{(\mathrm{R})} = \frac{\hat{\mathbf{p}}_{\mathrm{S}}^2}{2m_{\mathrm{S}}} + \sum_{n=0}^{\infty} \frac{m_{\mathrm{S}}}{n!} V^{(n)}(\hat{\mathbf{x}}_{\mathrm{S}})(-\hat{\mathbf{x}}_{\mathrm{M}})^n. \quad (27)$$

The last term is just the Taylor series of the potential $V(\hat{\mathbf{x}}_{\mathrm{S}} - \hat{\mathbf{x}}_{\mathrm{M}})$ around $\hat{\mathbf{x}}_{\mathrm{S}}$. Thus,

$$\hat{H}_{\mathrm{MS}}^{(\mathrm{R})} = \frac{\hat{\mathbf{p}}_{\mathrm{S}}^2}{2m_{\mathrm{S}}} + m_{\mathrm{S}} V(\hat{\mathbf{x}}_{\mathrm{S}} - \hat{\mathbf{x}}_{\mathrm{M}}). \quad (28)$$

Note that the dependence on the position of the massive object relative to R has changed from a definite value in $\hat{H}_{\mathrm{RS}}^{(\mathrm{M})}$ to an operator in $\hat{H}_{\mathrm{MS}}^{(\mathrm{R})}$. Thus, when acting on a state in which the masses are in a superposition, the potential will look different in each branch. With the Hamiltonian in the frame of R, we can now determine the evolution of a quantum state

$$|\psi\rangle_{\mathrm{RMS}}^{(\mathrm{R})} = |\mathbf{0}\rangle_{\mathrm{R}} \frac{1}{\sqrt{K}} \left( \sum_{i=1}^{K} \left| \mathbf{x}_{\mathrm{M}}^{(i)} \right\rangle_{\mathrm{M}} \right) |\phi\rangle_{\mathrm{S}}, \quad (29)$$

with a general quantum state $|\phi\rangle$ of the probe particle S directly as

$$|\psi(t)\rangle_{\mathrm{RMS}}^{(\mathrm{R})} = e^{-\frac{i}{\hbar}\hat{H}_{\mathrm{MS}}^{(\mathrm{R})}t} |\psi\rangle_{\mathrm{RMS}}^{(\mathrm{R})}. \quad (30)$$

Similarly, we can consider the Hamiltonian for $N$ masses,

$$\hat{H}_{\mathrm{RS}}^{(\mathrm{M})} = \frac{\hat{\boldsymbol{\pi}}_{\mathrm{S}}^2}{2m_{\mathrm{S}}} + m_{\mathrm{S}} V(|\hat{\mathbf{q}}_{\mathrm{S}} - \hat{\mathbf{q}}_{\mathrm{M}_i}|) \quad (31)$$

in the frame of M, where $V(|\hat{\mathbf{q}}_{\mathrm{S}} - \mathbf{q}_{\mathrm{M}_i}|) = -\sum_{i=1}^{N} \frac{GM_i}{|\hat{\mathbf{q}}_{\mathrm{S}} - \mathbf{q}_{\mathrm{M}_i}|}$ is the Newtonian potential, and we use that $\mathbf{q}_{\mathrm{M}_1} = 0$ and replace the vectors $\mathbf{q}_{\mathrm{M}_i}$ by the operators $\hat{\mathbf{q}}_{\mathrm{M}_i}$. This does not make a difference in which the positions of all masses are definite.

Applying the generalized transformation operator given in Eq. (12), we find that this Hamiltonian transforms as

$$\hat{H}_{\mathrm{MS}}^{(\mathrm{R})} = \left(\hat{S}^{\mathrm{R}\to\mathrm{M}}\right)^{\dagger} \hat{H}_{\mathrm{RS}}^{(\mathrm{M})} \hat{S}^{\mathrm{R}\to\mathrm{M}} = \frac{\hat{\mathbf{p}}_{\mathrm{S}}^2}{2m_{\mathrm{S}}} + m_{\mathrm{S}} V(|\hat{\mathbf{x}}_{\mathrm{S}} - \hat{\mathbf{x}}_{\mathrm{M}_i}|). \quad (32)$$

**Explicit calculation for the general case of $N$ masses.** As introduced in "Generalization to $N$ masses", the operator which performs the QRF change transformation from system R to system M is given by

$$\hat{S}^{\mathrm{R}\to\mathrm{M}} = \hat{T}_{\mathrm{rel}}^{(\mathrm{M})\dagger} \circ \hat{\mathcal{P}}_{\mathrm{MR}} \circ \hat{U}_{\mathrm{MSR}} \circ \hat{T}_{\mathrm{rel}}^{(\mathrm{M})}. \quad (33)$$

Let us take a closer look at the individual components of this operator. The first step is to change the coordinates of M to an origin, three distinguished axes and otherwise relative coordinates. The origin and the three axes are given by

$$\mathbf{x}_1 \to \mathbf{x}_1 \, (origin), \quad (34)$$

$$\mathbf{x}_2 \to \mathbf{x}_2 - \mathbf{x}_1 \equiv \mathbf{a}, \quad (35)$$

$$\mathbf{x}_3 \to \mathbf{x}_3 - \mathbf{x}_1 \equiv \mathbf{b}, \quad (36)$$

$$\mathbf{x}_4 \to \mathbf{x}_4 - \mathbf{x}_1 \equiv \mathbf{c}. \quad (37)$$

All other degrees of freedom of M are expressed relative to this frame. That is, for all $n \notin \{1, 2, 3, 4\}$, the relative position vector is decomposed as

$$\widetilde{\mathbf{x}}_n \equiv \mathbf{x}_n - \mathbf{x}_1 = r_n^1 \mathbf{a} + r_n^2 \mathbf{b} + r_n^3 \mathbf{c} \quad (38)$$

and each position vector is mapped as $\mathbf{x}_n \to \mathbf{r}_n$. By inverting the above relation, one can find an explicit expression of $r_n^1, r_n^2$ and $r_n^3$ in terms of the components of $\mathbf{x}_1, \mathbf{x}_2, \mathbf{x}_3, \mathbf{x}_4$, and $\mathbf{x}_n$. Altogether, we

thus have the following transformation:

$$
\begin{pmatrix} \mathbf{x}_1 \\ \mathbf{x}_2 \\ \mathbf{x}_3 \\ \mathbf{x}_4 \\ \mathbf{x}_5 \\ \vdots \\ \mathbf{x}_N \end{pmatrix} \rightarrow \begin{pmatrix} \mathbf{x}_1 \\ \mathbf{x}_2 - \mathbf{x}_1 \\ \mathbf{x}_3 - \mathbf{x}_1 \\ \mathbf{x}_4 - \mathbf{x}_1 \\ \mathbf{r}_5(\mathbf{x}_1, \mathbf{x}_2, \mathbf{x}_3, \mathbf{x}_4, \mathbf{x}_5) \\ \vdots \\ \mathbf{r}_N(\mathbf{x}_1, \mathbf{x}_2, \mathbf{x}_3, \mathbf{x}_4, \mathbf{x}_N) \end{pmatrix}.
\tag{39}
$$

The operator implementing this transformation is

$$
\hat{T}^{(M)}_{\text{rel}} = \int \left( \prod_{i=1}^{N} d^2 \mathbf{x}_i \right) \sqrt{|\det J|} |\mathbf{x}_1\rangle \langle \mathbf{x}_1|_{M_1} \otimes |\mathbf{a}(\mathbf{x}_1, \mathbf{x}_2)\rangle \langle \mathbf{x}_2|_{M_2}
$$
$$
\otimes |\mathbf{b}(\mathbf{x}_1, \mathbf{x}_3)\rangle \langle \mathbf{x}_3|_{M_3} \otimes |\mathbf{c}(\mathbf{x}_1, \mathbf{x}_4)\rangle \langle \mathbf{x}_4|_{M_4} \otimes
$$
$$
\bigotimes_{n=5}^{N} |\mathbf{r}_n(\mathbf{x}_1, \mathbf{x}_2, \mathbf{x}_3, \mathbf{x}_4, \mathbf{x}_n)\rangle \langle \mathbf{x}_n|_{M_n} \otimes \mathbb{1}_{\text{SR}},
\tag{40}
$$

where $J$ denotes the Jacobian of the transformation satisfying $\det J \neq 0$. The advantage of this strategy is that the relative coordinates $\mathbf{r}_5, \ldots, \mathbf{r}_N$ are invariant under the rotations and shifts and thus factorize out for the configurations considered here.

The next step of the QRF change is implemented by the operator

$$
\hat{U}_{\text{MSR}} = \int d^3 \mathbf{x}_1 d^3 \mathbf{a} \underbrace{|R(-\theta(\mathbf{e}_1, \mathbf{a}))\mathbf{x}_1\rangle \langle \mathbf{x}_1|_{M_1} \otimes |\mathbf{a}\rangle \langle \mathbf{a}|_{M_2}}_{\text{controls and rotates on part of M}}
$$
$$
\otimes \underbrace{\hat{R}_{M_3}(-\theta(\mathbf{e}_1, \mathbf{a})) \otimes \hat{R}_{M_4}(-\theta(\mathbf{e}_1, \mathbf{a})) \otimes \mathbb{1}_{M_5, \ldots, M_N}}_{\text{rotates rest of M}}
$$
$$
\otimes \underbrace{e^{\frac{i}{\hbar} R(-\theta(\mathbf{e}_1, \mathbf{a}))\mathbf{x}_1 \hat{P}_S} \hat{R}_S(-\theta(\mathbf{e}_1, \mathbf{a}))}_{\text{shifts and rotates S}} \otimes \underbrace{(\mathbb{1}_{R_1} \otimes e^{-\frac{i}{\hbar}(R(-\theta(\mathbf{e}_1, \mathbf{a}))\mathbf{a} - \mathbf{e}_1)\hat{P}_{R_2}})}_{\text{adjusts R}},
\tag{41}
$$

where $\hat{R}_J(-\theta(\mathbf{e}_1, \mathbf{a}))$ denotes the three-dimensional rotation matrix that rotates system $J = M_3, M_4, S$ by $-\theta(\mathbf{e}_1, \mathbf{a})$, with $\theta(\mathbf{e}_1, \mathbf{a})$ the angle between the vectors $\mathbf{e}_1$ connecting $R_1$ and $R_2$, and $\mathbf{a}$ connecting masses $M_1$ and $M_2$. In particular, the operator controls on systems $M_1$ and $M_2$ and reads out position $\mathbf{x}_1$ and relative distance $\mathbf{a}$. Systems $M_1, M_3$, and $M_4$ are then rotated accordingly, as can be seen in Fig. 3. Note that $M_2$ will be modified later by the operator $\hat{\mathcal{P}}_{\text{MR}}$. Likewise, system S is rotated and shifted by $-R(-\theta(\mathbf{e}_1, \mathbf{a}))\mathbf{x}_1$.

Recall that in three spatial dimensions, a rotation can be fully characterized by one angle $\theta$ plus an axis of rotation $\mathbf{u}$. For completeness, note that for general $\theta$ and normalized $\mathbf{u}$, the rotation matrix about axis $\mathbf{u}$ by angle $\theta$ takes the form

$$
R = \begin{pmatrix} \cos\theta + u_x^2(1-\cos\theta) & u_x u_y(1-\cos\theta) - u_z \sin\theta & u_x u_z(1-\cos\theta) + u_y \sin\theta \\ u_y u_x(1-\cos\theta) + u_z \sin\theta & \cos\theta + u_y^2(1-\cos\theta) & u_y u_z(1-\cos\theta) - u_x \sin\theta \\ u_z u_x(1-\cos\theta) - u_y \sin\theta & u_z u_y(1-\cos\theta) + u_x \sin\theta & \cos\theta + u_z^2(1-\cos\theta) \end{pmatrix}.
\tag{42}
$$

Thus, in order to rotate $\mathbf{a}$ into $\mathbf{e}_1$, we apply a rotation in the plane orthogonal to $\mathbf{u} = \frac{\mathbf{e}_1 \times \mathbf{a}}{|\mathbf{e}_1 \times \mathbf{a}|}$ by the angle $-\theta = -\arccos\frac{\mathbf{a} \cdot \mathbf{e}_1}{|\mathbf{a}|}$. The resulting rotation matrix is completely specified by the vector $\mathbf{a}$ and its relation to $\mathbf{e}_1$.

Finally, system R is adjusted such that the vector between $R_1$ and $R_2$ has the same length as $\mathbf{a}$. This is necessary since R is later swapped with M and we must avoid any transformation on system M that is not a relative-distance-preserving transformation, such as stretching or squeezing. Moreover, this change on $R_2$ does not affect the dynamics since its contribution in the Hamiltonian is trivial.

After this, the generalized parity swap operator

$$
\hat{\mathcal{P}}_{\text{MR}} = \text{SWAP}_{(M_i, R_i)_{i=1,2}} \circ \int d^3\mathbf{x} |-\mathbf{x}\rangle \langle \mathbf{x}|_{M_1} \otimes \int d^3\mathbf{a} |F_{\mathbf{e}_1}\mathbf{a}\rangle \langle \mathbf{a}|_{M_2} \otimes \mathbb{1}_{M_3 \ldots M_N \text{SR}}
\tag{43}
$$

is applied. This implements a reflection of $M_1$ about the origin and a reflection of $M_2$ about the $\mathbf{e}_1$-axis.

Finally, the labels of $M_1$ and $R_1$, and $M_2$ and $R_2$ are exchanged respectively. As a last step, we apply the inverse transformation $\hat{T}^{(M)\dagger}_{\text{rel}}$. At the end, we are left with the coordinates relative to the origin $M_1$ in system M and a right-handed orthonormal frame attached to $M_1$ and $M_2$. When applied to a state of the form (11), the end result is a state of the form (16), thereby changing into a quantum reference frame where the metric is definite. Applied to a quantum state with one single amplitude for the mass distribution, the QRF change operator $\hat{S}^{R \to M}$ reduces to a classical coordinate transformation, involving only translations and rotations in three-dimensional space.

**Relative-distance-preserving transformation using an auxiliary system.** As mentioned in "Generalization to $N$ masses", it is possible to implement QRF changes that go beyond quantum superpositions of Euclidean isometries. In this section, we present a more general QRF change operator $\hat{\mathcal{V}}$ that can implement more general relative-distance-preserving transformations. Note that this operator $\hat{\mathcal{V}}$ can perform general quantum coordinate changes. For the sake of the main argument made in this article, i.e. predicting the dynamics of a probe system S in the vicinity of mass configurations in quantum superpositions using QRF transformations and the dynamics in the frame of the massive objects, one does have to restrict to relative-distance-preserving transformations, though.

This more general QRF change operator requires an additional, auxiliary system which marks the branch of the superposition of the configuration. If the ancilla system is in a different, orthogonal state in each branch of the superposition, this allows to control on and perform different transformations in each branch. This may seem like too strong an assumption. However, some experimental set-ups such as spin-controlled interferometry do involve an additional system that is required to construct the superposition of mass configurations and that becomes entangled with the latter in the process[23].

In the following, we restrict to two point masses and two distinct mass configurations in superposition. It is straightforward to generalize to $N$ point masses and $K$ configurations in superposition. Consider the following state of an ancillary system A, two masses $M_1$ and $M_2$, and a probe system S:

$$
\frac{1}{\sqrt{2}} \left( |0\rangle_A |x_{1,0}^\mu\rangle_{M_1} |x_{2,0}^\mu\rangle_{M_2} + |1\rangle_A |x_{1,1}^\mu\rangle_{M_1} |x_{2,1}^\mu\rangle_{M_2} \right) \otimes |x_S^\mu\rangle_S.
\tag{44}
$$

Here, $|x^\mu\rangle$ denotes the quantum state of a four-vector $x^\mu \in \mathbb{R}^{(1,3)}$ such that $|x^\mu\rangle \in L^2(\mathbb{R}^{(1,3)})$. The total state can be seen as the one relative to some reference system R, whose state can be omitted since it is not required to distinguish the different branches. Note that the ancillary system takes on different orthogonal states in different branches. This allows us to implement different unitary operations in different branches while still keeping the unitarity of the total operator.

We are looking for a transformation $\hat{\mathcal{V}}$ that maps the state (44) to

$$
|\tilde{x}_1^\mu\rangle_{M_1} |\tilde{x}_2^\mu\rangle_{M_2} \otimes \frac{1}{\sqrt{2}} \left( |0\rangle_A |\tilde{x}_{S,0}^\mu\rangle_S + |1\rangle_A |\tilde{x}_{S,1}^\mu\rangle_S \right),
\tag{45}
$$

in which the states of the masses factorize out. Hence, in the new quantum coordinate system, the mass configuration is classical and thus also the spacetime sourced by the masses. Due to the principle of "covariance of dynamical laws under quantum coordinate transformations", the equations of motion retain their form under this quantum change of coordinates. The transformation $\hat{\mathcal{V}}$ is a quantum-controlled unitary of the form

$$\hat{\mathcal{V}} = |0\rangle\langle 0|_A \otimes (\hat{U}_0)_{M_1,M_2,S} + |1\rangle\langle 1|_A \otimes (\hat{U}_1)_{M_1,M_2,S}, \tag{46}$$

such that

$$\hat{U}_0 \left|x^\mu_{1,0}\right\rangle_{M_1} \left|x^\mu_{2,0}\right\rangle_{M_2} \left|x^\mu_S\right\rangle_S = \left|\tilde{x}^\mu_1\right\rangle_{M_1} \left|\tilde{x}^\mu_2\right\rangle_{M_2} \left|\tilde{x}^\mu_{S,0}\right\rangle_S, \tag{47}$$

$$\hat{U}_1 \left|x^\mu_{1,1}\right\rangle_{M_1} \left|x^\mu_{2,1}\right\rangle_{M_2} \left|x^\mu_S\right\rangle_S = \left|\tilde{x}^\mu_1\right\rangle_{M_1} \left|\tilde{x}^\mu_2\right\rangle_{M_2} \left|\tilde{x}^\mu_{S,1}\right\rangle_S. \tag{48}$$

Here, $\tilde{x}(x)$ is a "quantum coordinate system", meaning it assigns different coordinates to all systems in different branches of the superposition. Hence, the coordinates are implicitly given with respect to a reference frame that is itself in a superposition relative to the old reference frame. The general form of the unitary operations $\hat{U}_0$ and $\hat{U}_1$ is given by

$$\hat{U}_0 = \int d^4 x_1 \sqrt{|\det J_0|} |f^\mu_0(x^\mu_1)\rangle\langle x^\mu_1|_{M_1} \otimes \int d^4 x_2 \sqrt{|\det J_0|} |f^\mu_0(x^\mu_2)\rangle\langle x^\mu_2|_{M_2}$$
$$\otimes \int d^4 x_S \sqrt{|\det J_0|} |f^\mu_0(x^\mu_S)\rangle\langle x^\mu_S|_S, \tag{49}$$

$$\hat{U}_1 = \int d^4 x_1 \sqrt{|\det J_1|} |f^\mu_1(x^\mu_1)\rangle\langle x^\mu_1|_{M_1} \otimes \int d^4 x_2 \sqrt{|\det J_1|} |f^\mu_1(x^\mu_2)\rangle\langle x^\mu_2|_{M_2}$$
$$\otimes \int d^4 x_S \sqrt{|\det J_1|} |f^\mu_1(x^\mu_S)\rangle\langle x^\mu_S|_S, \tag{50}$$

where $\det J_{0/1} \neq 0$. Here, we set the functions $f^\mu_{0/1}$ such that

$$f^\mu_0\left(x^\mu_{1,0}\right) \equiv f^\mu_1\left(x^\mu_{1,1}\right) =: \tilde{x}^\mu_1, \tag{51}$$

$$f^\mu_0\left(x^\mu_{2,0}\right) \equiv f^\mu_1\left(x^\mu_{2,1}\right) =: \tilde{x}^\mu_2. \tag{52}$$

The operators $\hat{U}_{0/1}$ each perform a classical coordinate transformation $x^\mu \to f_{0/1}(x^\mu)$. Now, in the coordinate system $\tilde{x}(x)$, the masses are in definite positions and source a classical spacetime. We can then solve the geodesic equation separately for the two initial conditions $\tilde{x}^\mu_{S,0} \equiv \tilde{x}^\mu_{S,0}(\tau = 0)$ and $\tilde{x}^\mu_{S,1} \equiv \tilde{x}^\mu_{S,1}(\tau = 0)$, where $x(\tau)$ and $t(\tau)$ denote the coordinate position and coordinate time, and $\tau$ is the parameter in the geodesic equation. Furthermore, the phase between these two different semi-classical paths can be determined using Eq. (5). Evolving the states in time, this gives rise to the state

$$\left|\tilde{x}^\mu_1\right\rangle_{M_1} \left|\tilde{x}^\mu_2\right\rangle_{M_2} \otimes \frac{1}{\sqrt{2}} \left(e^{-\frac{i}{\hbar}\Phi^{(0)}} |0\rangle_A \left|\tilde{x}^\mu_{S,0}(\tau)\right\rangle_S + e^{-\frac{i}{\hbar}\Phi^{(1)}} |1\rangle_A \left|\tilde{x}^\mu_{S,1}(\tau)\right\rangle_S\right), \tag{53}$$

where $\Phi^{(i)}$ is defined through Eq. (5). Note that we assume the masses $M_1$ and $M_2$ to remain static during the entire evolution. As mentioned earlier, this assumption is justified in case of experimental set-ups in which the masses are trapped and therefore stationary. Now, we can apply the inverse

transformation $\hat{\mathcal{V}}^\dagger$ to the global state:

$$\hat{\mathcal{V}}^\dagger \left(\left|\tilde{x}^\mu_1\right\rangle_{M_1} \left|\tilde{x}^\mu_2\right\rangle_{M_2} \otimes \frac{1}{\sqrt{2}}\left(e^{-\frac{i}{\hbar}\Phi^{(0)}}|0\rangle_A \left|\tilde{x}^\mu_{S,0}(\tau)\right\rangle_S + e^{-\frac{i}{\hbar}\Phi^{(1)}}|1\rangle_A \left|\tilde{x}^\mu_{S,1}(\tau)\right\rangle_S\right)\right)$$
$$= \frac{1}{\sqrt{2}}\left(e^{-\frac{i}{\hbar}\Phi^{(0)}}|0\rangle_A \hat{U}_0^\dagger\left(\left|\tilde{x}^\mu_1\right\rangle_{M_1}\left|\tilde{x}^\mu_2\right\rangle_{M_2}\left|\tilde{x}^\mu_{S,0}(\tau)\right\rangle_S\right) + e^{-\frac{i}{\hbar}\Phi^{(1)}}|1\rangle_A \hat{U}_1^\dagger\left(\left|\tilde{x}^\mu_1\right\rangle_{M_1}\left|\tilde{x}^\mu_2\right\rangle_{M_2}\left|\tilde{x}^\mu_{S,1}(\tau)\right\rangle_S\right)\right)$$
$$= \frac{1}{\sqrt{2}}\left(e^{-\frac{i}{\hbar}\Phi^{(0)}}|0\rangle_A \left|f^{-1}_0(\tilde{x}^\mu_1)\right\rangle_{M_1}\left|f^{-1}_0(\tilde{x}^\mu_2)\right\rangle_{M_2}\left|f^{-1}_0(\tilde{x}^\mu_{S,0}(\tau))\right\rangle_S\right.$$
$$\left. + e^{-\frac{i}{\hbar}\Phi^{(1)}}|1\rangle_A \left|f^{-1}_1(\tilde{x}^\mu_1)\right\rangle_{M_1}\left|f^{-1}_1(\tilde{x}^\mu_2)\right\rangle_{M_2}\left|f^{-1}_1(\tilde{x}^\mu_{S,1}(\tau))\right\rangle_S\right). \tag{54}$$

Thus, relative to the initial frame of reference, the probe system S becomes entangled with the positions of the masses during its time evolution. In each separate branch $(i)$, $i = 0, 1$, S follows a geodesic in the spacetime that would be sourced if the masses were completely classical and at positions $x^\mu_{1,i}$ and $x^\mu_{2,i}$.

One option to verify these predictions is to disentangle the ancilla system from the rest of the systems by post-selecting in the basis $\{\frac{1}{\sqrt{2}}(|0\rangle + |1\rangle), \frac{1}{\sqrt{2}}(|0\rangle - |1\rangle)\}$. Then, conditioned on the result observed, one can certify the entanglement of the probe system with the masses[15,16] and thus corroborate the predictions made based on the extended principle of covariance.

## Acknowledgements
We thank Markus Aspelmeyer, Luis C. Barbado, Carlo Cepollaro, Marios Christodoulou, Flaminia Giacomini, and Wolfgang Wieland for stimulating discussions and helpful comments on an earlier version of this work. E.C.-R. is supported by an ETH Zurich Postdoctoral Fellowship and acknowledges financial support from the Swiss National Science Foundation (SNSF) via the National Centers of Competence in Research QSIT and SwissMAP, as well as the project No. 200021_188541. We acknowledge financial support by the Austrian Science Fund (FWF) through BeyondC (F7103-N48), the Austrian Academy of Sciences (ÖAW) through the project "Quantum Reference Frames for Quantum Fields" (ref. IF 2019 59 QRFQF), the European Commission via Testing the Large-Scale Limit of Quantum Mechanics (TEQ) (No. 766900) project, the Foundational Questions Institute (FQXi) and the Austrian-Serbian bilateral scientific cooperation no. 451-03-02141/2017-09/02. This publication was made possible through the support of the ID 61466 grant from the John Templeton Foundation, as part of The Quantum Information Structure of Spacetime (QISS) Project (qiss.fr). The opinions expressed in this publication are those of the authors and do not necessarily reflect the views of the John Templeton Foundation.

## Author contributions
A.C.dlH., V.K., E.C.R., and Č.B. contributed to all aspects of the research, with leading and equal input from A.C.dlH. and V.K.

## Competing interests
The authors declare no competing interests.

## Data availability
Data sharing not applicable to this article as no datasets were generated or analyzed during the current study.

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
