## [Peer Review File · Communications Physics]

REVIEWER COMMENTS

Reviewer #1 (Remarks to the Author: Overall significance):

See attached PDF

Reviewer #1 (Remarks to the Author: Impact):

See attached PDF

Reviewer #1 (Remarks to the Author: Strength of the claims):

See attached PDF

Reviewer #1 (Remarks to the Author: Reproducibility):

See attached PDF

Reviewer #2 (Remarks to the Author: Overall significance):

In the present manuscript, the authors propose a strategy to determine the dynamics of objects in the presence of mass configurations in superposition. They provide a argument that allows to make predictions where the gravitational source is not in a classical configuration but in a quantum superposition. They found that the frame can be changed associated with the massive object by using the QRF transformation. It is interesting to see that the gravitational field is definite by employing the QRF transformation, and the time evolution of objects can be determined in the presence of the mass configuration in the performed model. It is shown that the approach enables to make the spacetime globally definite and is thus restricted to a superposition of configurations related by relative distance preserving transformations. Overall, I think the topic of the work is interesting and significant therefore recommended publication.

Reviewer #2 (Remarks to the Author: Strength of the claims):

I think this work is convincing.

Reviewer #2 (Remarks to the Author: Reproducibility):

(1)The prerequisite for using QRF reference frame transformation is that the mass configurations in different branches are related by relatively stable transformation. What is the relatively stable transformation between the mass configurations in different branches?

(2)In the 4st paragraph of sec. I, the authors claimed that “The existence of test particles breaks the equivalence of the mass configurations related by these transformations in different branches”. I would

like to know how this inequality is dealt with in the later calculations.

(3)How to explain all the masses are in a definite position for all amplitudes in the bottom right and bottom left of Fig.4, and what does represent in the Fig.4?

(4)In the text before Eq. (32), the author applying the generalized transformation operator given in Eq. (33). I suggest that Eq. (33) be changed to Eq. (12).

I have been asked to review the manuscript titled "Falling through masses in superposition...". In this paper the authors propose "a strategy to determine the dynamics of objects in the presence of mass configurations in superposition, and hence an indefinite spacetime metric, using quantum reference frame (QRF) transformations."

Recommendation

I believe that research in this area is of current interest. However, I believe that this work is fundamentally flawed and therefore recommend against publication. My comments can be viewed below.

English quality

The quality of the English is good. Below some small remarks.

- Change "lacking a theory which permits" to "lacking a theory that permits";
- Change "scenarios but in fact, they are inherent" to "scenarios, but in fact they are inherent";
- Below (1) you mention the Einstein tensor but do not define it explicitly (as the left hand of (1), but not all readers might know that);

Minor issues

Here I list minor issues.

- The authors say "mass configuration becomes definite." What is a "definite" mass configuration?
- The authors map the state (2) to the state (3) through their particular QRF transformation. I would assume that, if the state is to represent the same physical state, the QRF-transformed time-evolved initial state in frame R is equivalent to the time-evolved QRF-transformed initial state in frame M. Is this the case? It seems to me that the authors claim it is by making the final "inverse QRF transformation". This, I understand, is an Ansatz of the authors.

Major issues

Here I list major issues.

- In Section II.B the authors start talking about a mass M in the superposition of two locations. They say "This is motivated by most experimental proposals ... magnetic traps". They refer to Figure 1, where they draw the Earth as M. Clearly, M the Earth is not localised by external traps but by its own gravitational field, which already begs the question of how it would be in superposition since it is classical and well localized. While I do understand that Figure 1 is just a pictorial representation, it is misleading.
- If the authors use the QRF to switch as depicted in Figure 1, what about the gravitational field and the forces induced by the traps (assuming the authors correct for the fact that one cannot trap the Earth)? Are the traps in a superposition as well? What about their gravitational field? I ask this because the mass M that the authors actually consider is that of a trapped massive, yet perhaps mesoscopic and not planet-like, object M. Figure 1 is thus quite misleading, because the traps that would confine the mass M in two locations are not in superposition, while the mass M is, but the authors also argue that the locations can be determined by the traps before the superposition is created, c.f "The spatial coordinates assigned to the massive object ... before the mass is placed in the traps".
- I have a question about (2). In order to separate the Hilbert space, a quantisation procedure must be determined. This, in turn, basically requires us to have a Hamiltonian. In order to have a Hamiltonian we have to be able to define it, which is a major problem when gravity is not weak and dynamical. How can the authors ensure that (2) is viable at all times, together with the quantisation procedure outlined later on in the manuscript?
- The authors say "Assuming that we can associate a semi-classical state $|g\rangle$ to the gravitational field

sourced by the mass M ...". However, the "different branches" of the mass M will start interacting at some point in time and therefore, as expected by overwhelming evidence in Nature, to decohere to some "classical state". Even if one were, somehow, to assume that the mass M is light enough, and therefore its gravitational field is weak enough, the point is that the authors compute the time evolution of system in the frame where M is defined (using their language), and they do so to lowest order in the gravitational coupling (i.e., \sqrt{GM}). This means that the mass M in the different branches has not yet interacted, which is possible only if the probe S is in the middle region between the two branches and it evolves for a time shorter than $|x_m^{(1)} - x_m^{(2)}|/c$. After that, the gravitons exchanged between the branches will make the whole scheme fail to predict. If this were not the case, we would expect to see superposition of macroscopic objects. Note that if the distance between S and one of the branches of M is larger than that of the two branches itself, I am convinced that the premises on which the predictions of this work are obtained are just wrong. In fact, the branches of M would start interacting before they even affect the system and therefore the whole scheme is not predictive.

- This begs another consideration: if the two branches do not interact gravitationally, because the R -frame scenario is equivalent to the M -frame scenario, then one would conclude that the mass M is not gravitating, somehow. I hope that the authors understand this point. If, as the gravitational collapse models claim, the different branches interact, then how can you move to a frame where the mass M is definite and therefore does not self interact (at least in the regime considered here)? I do believe that Nature shows that superpositions of macroscopic objects cannot survive long, and I hope that the authors will agree on this.

The authors recognise, in the end, that their principle of "covariance of dynamical laws under quantum coordinate transformations" must be tested. However, how can they reconcile it with the fact that we do not see many superpositions of large masses? I fear that this argument already invalidates the possibility of the existence of a "covariance of dynamical laws under quantum coordinate transformations".

- I add a note on my previous comments. It could, and I emphasise 'could', be that if there is only one mass M in superposition and a probe mass S in the Universe, then somehow the supposition holds for long times. I am not aware of any theory that predicts that, and the authors anyway assume lowest order interactions. Thus, I am quite convinced that my criticism outlined above cannot be invalidated because higher order computations are missing. I fear that these cannot be done anyway for technical and foundational reasons.

- The authors say "While it is possible to reproduce this effect in a definite spacetime if the clock is set to follow a superposition of two different trajectories, this is impossible for a clock following a single trajectory as we consider here.". As after as I understand this claim defeats the whole point of their paper. The authors invoke the "covariance of dynamical laws under quantum coordinate transformations" to map the situation of a mass M in superposition to a case of a mass M in a definite state. The results should be independent of the situation in which the authors choose to do the calculations. If this were not the case, then the whole claim of "covariance of dynamical laws under quantum coordinate transformations" would fail. Thus, I conclude that while it is possible to reproduce this effect in a definite spacetime if the clock is set to follow a superposition of two different trajectories, this is impossible for a clock following a single trajectory as we consider here." Is wrong. To put it differently: superposed M and localized moving S is equivalent to superposed moving S and localized M , according to the very "covariance of dynamical laws under quantum coordinate transformations".

- Why is q_M classical in (23). How can you "sit" on the mass M to give it a classical definite $q_M=0$ coordinates but still assume that it is a quantum object? If M can be put into superpositions, I would assume that one cannot attach to it a reference frame where it has $q_M=0$. Definite zero (classical?) position means complete lack of knowledge of the momentum of the object. With unknown repercussions on its gravitational field, since gravity depends on the energy...

Once could say that M is in the frame where it has average null position, but then I would still expect q_M to have a hat and thus be an operator, and thus the sentence starting with "Note that the dependence on the position..." to be incorrect.

- Reading... this paper, I am coming to the conclusion that the QRF transformations apply only in the case of weak gravitational fields. In fact, it is completely unclear to me what it would even mean to have a frame

where the source of gravity is in a superposition if it is not possible to localise it in the first place, nor to give it a definite Hilbert space.

- Finally, regarding the “covariance of dynamical laws under quantum coordinate transformations”, I do not understand how they could work (assuming they do) beyond the case where the source of gravity does not self interact, in contradiction with what we expect. In fact, In frame R the mass M is in superposition and we expect its state to decohere somehow. In frame M, it is in a definite state and it should not decohere. Thus, will it decohere or not? I cannot believe that the process of decoherence is frame dependent. This leads me to conclude, as noted above, that this work could be, at best, predictive for timescales shorter than those when self decoherence of a mass in a nonclassical begins to take place.

- Given the above, the claims of generality in the Discussion are over the top.

- The claim in “However, we go beyond the ... being related by a perturbative contribution.” Is incorrect. The ability to define a definite Hilbert space for the gravitating mass can be done, in the simple way that the authors implement their transformations, only in the linearised regime. Beyond that, I understand that it is even unclear how to univocally assign Hamiltonians to strongly gravitating and dynamical quantum systems, and to determine their position.

Overall comment

Given my comments above, it is clear that this work might be valid, if any, only when linearising general relativity and working to lowest order. In this sense, even if the authors were to reply to all of my criticisms and convince me that they are right, or fix the parts where the issues that I raise are indeed valid, it is not clear what statements about gravity are left. Spacetime, in this work, is Minkowski and the linearisation and first order regime means, de facto, that the authors are ignoring all of the interesting (e.g., nonlinear) aspects of gravity and working in a special relativistic setting with some Newtonian potentials where the sources of gravity do not have time to self-interact. While this can be interesting, and I think it is, it also means that the work is oversold.

In addition, I would like to make the point that the authors cannot get away from this by just saying that one can easily extend their work to nonlinear and strong gravity, and that this will be done in later work. Either it is clear (theoretically and formally, not by words) already here that everything does carry through to strong gravity/nonlinear regimes, or the claim about adding insight on the overlap of general relativity and quantum mechanics loses support.

References

The authors should add **at least** the following references which, I understand, are not directly relevant to the calculations but are for sure relevant for the broad context:

[New Journal of Physics 21, 043047 (2019)]

[Physics Letters B 54, 182-186 (2016)]

I also take the opportunity to note that (some of) the authors, in the recent years, tend to ignore - for whatever reason - the work of colleagues (among others) that authored the papers suggested here. I would recommend the authors to be more “inclusive” (a buzzword invoked on a hourly basis nowadays) in the future.

Final comment.

I am aware of the fact that my report is quite critical, and I do believe that this work is fundamentally flawed. Nevertheless, while I stand against publication, I am open to be proven wrong. I do not see how this could happen, but nevertheless I am can potentially change my opinion. I wished to make this clear with the authors, who might otherwise have the impression that I just do not like their work.

In the present manuscript, the authors propose a strategy to determine the dynamics of objects in the presence of mass configurations in superposition. They provide an argument that allows to make predictions where the gravitational source is not in a classical configuration but in a quantum superposition. They found that the frame can be changed associated with the massive object by using the QRF transformation. It is interesting to see that the gravitational field is definite by employing the QRF transformation, and the time evolution of objects can be determined in the presence of the mass configuration in the performed model. It is shown that the approach enables to make the spacetime globally definite and is thus restricted to a superposition of configurations related by relative distance preserving transformations. Overall, I think the topic of the work is interesting and significant therefore recommended publication. However, there are several issues required to be addressed.

(1) The prerequisite for using QRF reference frame transformation is that the mass configurations in different branches are related by relatively stable transformation. What is the relatively stable transformation between the mass configurations in different branches?

(2) In the 4th paragraph of sec. I, the authors claimed that “The existence of test particles breaks the equivalence of the mass configurations related by these transformations in different branches”. I would like to know how this inequality is dealt with in the later calculations.

(3) How to explain all the masses are in a definite position for all amplitudes in the bottom right and bottom left of Fig.4, and what does f_1 represent in the Fig.4?

(4) In the text before Eq. (32), the author applying the generalized transformation operator given in Eq. (33). I suggest that Eq. (33) be changed to Eq. (12).

To sum up, due to its originality, I think the paper can be considered for publication if the above issues are dealt with completely.

The authors provide an argument that allows to make predictions where the gravitational source is not in a classical configuration but in a quantum superposition.

They assumed the covariance of dynamical laws under quantum coordinate transformations and applied the inverse transformation which yields the dynamical description in the original frame. It is found that the frame can be changed associated with the massive object by using the QRF transformation. In the present framework, the gravitational field is definite, and the concrete problems can be solved. In addition, the time evolution of objects can be determined in the presence of the mass configuration by using the performed model.

Author Response to Reviews of

Falling through masses in superposition: quantum reference frames for indefinite metrics

A.-C. de la Hamette, V. Kabel, E. Castro-Ruiz, Č. Brukner
GUIDEDOA-22-00431

Response to Reviewer #1

We would like to begin by thanking the referee for their efforts in carefully assessing our work. We studied the comments point by point and hope to have adequately addressed all criticisms raised in the report.

English quality

RC: *The quality of the English is good. Below some small remarks.*

- *Change “lacking a theory which permits” to “lacking a theory that permits”;*
- *Change “scenarios but in fact, they are inherent” to “scenarios, but in fact they are inherent”;*
- *Below (1) you mention the Einstein tensor but do not define it explicitly (as the left hand of (1), but not all readers might know that);*

AR: *We changed these points in the manuscript.*

Minor issues

RC: *Here I list minor issues. The authors say “mass configuration becomes definite.” What is a “definite” mass configuration?*

AR: *A definite mass configuration is one in which each constituent of the gravitational source can be assigned a single position and hence the gravitational field sourced by it is classical. We included this clarification in the introduction.*

RC: *The authors map the state (2) to the state (3) through their particular QRF transformation. I would assume that, if the state is to represent the same physical state, the QRF-transformed time-evolved initial state in frame R is equivalent to the time-evolved QRF-transformed initial state in frame M. Is this the case? It seems to me that the authors claim it is by making the final “inverse QRF transformation”. This, I understand, is an Ansatz of the authors.*

AR: *This is correct. It follows directly from the main assumption of our paper, that is covariance of dynamical laws under quantum coordinate transformations. See, for example, the sentence above Equation (10).*

RC: *Addressing the above points should improve the clarity of the paper.*

Major issues

- RC:** *Here I list major issues. In Section II.B the authors start talking about a mass M in the superposition of two locations. They say “This is motivated by most experimental proposals . . . magnetic traps”. They refer to Figure 1, where they draw the Earth as M . Clearly, M the Earth is not localised by external traps but by its own gravitational field, which already begs the question of how it would be in superposition since it is classical and well localized. While I do understand that Figure 1 is just a pictorial representation, it is misleading.*
- AR:** *This pictorial representation was chosen to illustrate that we are dealing with a massive object and to stay consistent with illustrations in previous work on superpositions of gravitational fields (see, e.g., “Zych, M., Costa, F., Pikovski, I. et al. Bell’s theorem for temporal order. Nat Commun 10, 3772 (2019).”). While we agree, of course, that putting the Earth in a quantum superposition of positions is practically impossible, we believe it is important to point out that neither quantum mechanics nor general relativity prevent us from performing such an experiment in principle. From this point of view, Fig. 1 is a sound qualitative depiction of the thought experiment. In Section II.D, we indicate concrete orders of magnitude for the various parameters that may be achievable in future experiments. In particular, we make it clear that experimental realizations would involve mesoscopic objects of the order of the Planck mass. We thus believe that it is clear to the reader that Figure 1 should be taken as a simplified, more pictorial representation of the situations we are considering throughout the paper.*
- RC:** *If the authors use the QRF to switch as depicted in Figure 1, what about the gravitational field and the forces induced by the traps (assuming the authors correct for the fact that one cannot trap the Earth)? Are the traps in a superposition as well? What about their gravitational field? I ask this because the mass M that the authors actually consider is that of a trapped massive, yet perhaps mesoscopic and not planet-like, object M . Figure 1 is thus quite misleading, because the traps that would confine the mass M in two locations are not in superposition, while the mass M is, but the authors also argue that the locations can be determined by the traps before the superposition is created, c.f. “The spatial coordinates assigned to the massive object . . . before the mass is placed in the traps”.*
- AR:** *Again, Figure 1 is hugely simplified, which is why the traps are not depicted. In a concrete experimental realization, one could for instance use optical or magnetic traps with potentials in superposition. However, importantly, using traps in a superposition is not the only possibility for realizing the experiment. One could, for instance, prepare a localized trap for every position of the mass in a superposition. Furthermore, experimental methods are proposed to prepare massive objects in superpositions without the need to trap them by post-selection of freely evolving matter waves [O. Romero-Isart, Quantum superposition of massive objects and collapse models, Physical Review A 84, 052121 (2011)]. Indeed, in any experimental realization, one has to be very careful about any gravitational fields sourced by objects other than the gravitational source under consideration. This includes the traps, other equipment, as well as any noise source. While this is an actual significant challenge for experimentalists (and currently a large focus of research in that area), it is common to neglect such additional sources when devising new theoretical strategies to deal with novel setups.*
- RC:** *I have a question about (2). In order to separate the Hilbert space, a quantisation procedure must be determined. This, in turn, basically requires us to have a Hamiltonian. In order to have a Hamiltonian we have to be able to define it, which is a major problem when gravity is not weak and dynamical. How can the authors ensure that (2) is viable at all times, together with the quantisation procedure outlined later on in the manuscript?*
- AR:** *We find this argument hard to follow. All that seems necessary at this stage is to have a Hilbert space (vector*

space equipped with inner product) and a partition into subsystems. It is common practice to define the latter at the kinematical level. The dynamics can then be added later. In fact, the aim of our work is to infer the dynamics (and thus the Hamiltonian) in presence of superpositions of gravitational sources through the argument presented. We move into the frame of reference where we have a definite mass configuration and where we can use standard Hamiltonian dynamics.

RC: *The authors say “Assuming that we can associate a semi-classical state $|g\rangle$ to the gravitational field sourced by the mass M ...”. However, the “different branches” of the mass M will start interacting at some point in time and therefore, as expected by overwhelming evidence in Nature, to decohere to some “classical state”. Even if one were, somehow, to assume that the mass M is light enough, and therefore its gravitational field is weak enough, the point is that the authors compute the time evolution of system in the frame where M is defined (using their language), and they do so to lowest order in the gravitational coupling (i.e., \sqrt{GM}). This means that the mass M in the different branches has not yet interacted, which is possible only if the probe S is in the middle region between the two branches and it evolves for a time shorter than $|x_m^{(1)} - x_m^{(2)}|/c$. After that, the gravitons exchanged between the branches will make the whole scheme fail to predict. If this were not the case, we would expect to see superposition of macroscopic objects. Note that if the distance between S and one of the branches of M is larger than that of the two branches itself, I am convinced that the premises on which the predictions of this work are obtained are just wrong. In fact, the branches of M would start interacting before they even affect the system and therefore the whole scheme is not predictive.*

AR: *Firstly, let us note that the statement “Assuming that we can associate a semi-classical state $|g\rangle$ to the gravitational field sourced by the mass M ...” is only made when comparing our results to those of Ref. [18], in which this assumption is made. None of our arguments rely on it; we are just mentioning it as we obtain the same results as these authors.*

As for the remainder of the argument, we find the referee’s comment that the branches of the mass will start interacting rather surprising. Naturally assuming they do in the context of our paper reveals a significant misunderstanding of our work. As the assumption of interacting branches is an important part of the referee’s general assessment, we would like to make our standing point very clear. Let us first make a general comment: standard quantum mechanics is a linear theory. Linear and unitary evolution means that different branches can interfere but do not interact with each other. A similar observation holds for quantum field theory. For example, when considering an analogous setup in QED, one finds that the different Feynman diagrams lead to interference terms but there are no photons exchanged between the various Feynman diagrams.

Now, we are aware that gravitational collapse models propose a gravity-induced interaction between the branches. However, these proposals are far from being universally accepted by the scientific community. It seems to us that the referee takes their validity for granted. On the other hand, in our work, we do not put in any assumption on whether individual quantum branches interact or whether they exchange gravitons. In fact, making any such assumption comes back to picking a specific approach to quantum gravity. This would run counter to the premise of our work, which is to remain agnostic about the quantum nature of the gravitational field. After making the argument, we find that the referee’s assumption would in fact lead to predictions in contradiction with the generalized principle of covariance (see Discussion in the paper). Therefore, we disagree with the premise that different branches will start interacting with one another through gravity. While we would not regard the interaction between the two branches as a priori impossible in gravity – indeed, it is our intention to contrast this view with others in our work – we are puzzled by the way the referee argues as if this is something that is universally accepted.

RC: *This begs another consideration: if the two branches do not interact gravitationally, because the R-frame scenario is equivalent to the M-frame scenario, then one would conclude that the mass M is not gravitating,*

somehow. I hope that the authors understand this point. If, as the gravitational collapse models claim, the different branches interact, then how can you move to a frame where the mass M is definite and therefore does not self interact (at least in the regime considered here)? I do believe that Nature shows that superpositions of macroscopic objects cannot survive long, and I hope that the authors will agree on this. The authors recognise, in the end, that their principle of “covariance of dynamical laws under quantum coordinate transformations” must be tested. However, how can they reconcile it with the fact that we do not see many superpositions of large masses? I fear that this argument already invalidates the possibility of the existence of a “covariance of dynamical laws under quantum coordinate transformations”.

AR: *We do not fully grasp the first part of this comment. The argument seems to be that if the branches do not interact gravitationally, then the mass cannot be gravitating. Again invoking the analogy to electrodynamics, this seems implausible - given an electron moving in a superposition of two different paths, we do not observe any interaction between the two branches. If this were the case, a single electron would repel itself as it approaches a diffraction grating - an effect that not only does not occur in experiments, but defies any known physical mechanics. Yet we would not conclude from the absence of interaction between the two branches that the electron cannot generate an EM field.*

Potentially, the argument is based on additional assumptions from the gravitational collapse models. In this case, it is no surprise that it runs counter to our conclusions since our results go against the predictions made using gravitational collapse models. This is clearly stated in the Discussion section.

Moreover, regarding the claim that “Nature shows that superpositions of macroscopic objects cannot survive long”: We agree that it is challenging to uphold the superposition of larger objects for long times due to decoherence, explainable within quantum theory. This is true for any quantum mechanical scenario, independently of whether it involves gravity or not. However, if environmental effects can be shielded well enough, it is possible to uphold the superposition for long enough times to study interesting effects (see recent advances and efforts in creating superpositions of objects of increasing size and mass). The fact that these decoherence effects do not show up in our calculations is because we rely on a simplified model that neglects the environmental effects that cause decoherence.

RC: ***I add a note on my previous comments. It could, and I emphasise ‘could’, be that if there is only one mass M in superposition and a probe mass S in the Universe, then somehow the supposition holds for long times. I am not aware of any theory that predicts that, and the authors anyway assume lowest order interactions. Thus, I am quite convinced that my criticism outlined above cannot be invalidated because higher order computations are missing. I fear that these cannot be done anyway for technical and foundational reasons.***

AR: *This is addressed by the reply to the comment above. Regarding the comment on whether or not massive objects in superposition can exist at all and are described by an existing theory: we can observe superposition states of objects that carry mass (e.g. electrons, Fullerene, biomolecules with 10 000 amu) and these are described by standard quantum mechanics. Of course these masses are rather small and in particular too small to give rise to observable gravitational effects. But we believe that by pushing the experimental capabilities, it should be possible to create a superposition of objects of larger mass (e.g. Schrödinger cat states). Although the breakdown of quantum theory is a logical possibility that we consider as an outcome of these experiments, it is certainly not the basis of our analysis. In fact, as discussed in the manuscript, such a breakdown would be contrary to our quantum symmetry principle, which is the most important assumption of our work.*

We are unsure why it is an issue that we assume lowest order interactions and what precisely is meant by that statement in the first place. If it is lowest order in the sense of quantum field theory, it seems necessary to assume a quantum field theoretic description of gravity to even make this statement. It is such a theory that we want to remain agnostic about.

RC: *The authors say “While it is possible to reproduce this effect in a definite spacetime if the clock is set to follow a superposition of two different trajectories, this is impossible for a clock following a single trajectory as we consider here.” As after as I understand this claim defeats the whole point of their paper. The authors invoke the “covariance of dynamical laws under quantum coordinate transformations” to map the situation of a mass M in superposition to a case of a mass M in a definite state. The results should be independent of the situation in which the authors choose to do the calculations. If this were not the case, then the whole claim of “covariance of dynamical laws under quantum coordinate transformations” would fail. Thus, I conclude that while it is possible to reproduce this effect in a definite spacetime if the clock is set to follow a superposition of two different trajectories, this is impossible for a clock following a single trajectory as we consider here.” Is wrong. To put it differently: superposed M and localized moving S is equivalent to superposed moving S and localized M , according to the very “covariance of dynamical laws under quantum coordinate transformations”.*

AR: *This seems to be a misunderstanding of our claims. We completely agree with the characterization in the last sentence: superposed M and localized moving S is equivalent to superposed moving S and localized M , according to the very “covariance of dynamical laws under quantum coordinate transformations”. However, what we mean in the paragraph alluded to is that if we have a single spacetime and a single trajectory, one cannot produce a superposition of different ticking rates in the clock. Thus, if one is to observe a superposition of ticking rates of a clock following a single trajectory, then one would have to conclude that this can only be due to spacetime in superposition. In order to be certain that the effect relies solely on gravitational time dilation, one should ideally repeat this experiment with different types of clocks. Invoking the universality of gravitational time dilation, one can then ensure that this is a purely gravitational effect and thus verify the superposition of geometries.*

RC: *Why is q_M classical in (23). How can you “sit” on the mass M to give it a classical definite $q_M = 0$ coordinates but still assume that it is a quantum object? If M can be put into superpositions, I would assume that one cannot attach to it a reference frame where it has $q_M = 0$. Definite zero (classical?) position means complete lack of knowledge of the momentum of the object. With unknown repercussions on its gravitational field, since gravity depends on the energy... Once could say that M is in the frame where it has average null position, but then I would still expect q_M to have a hat and thus be an operator, and thus the sentence starting with “Note that the dependence on the position... ” to be incorrect.*

AR: *In Section II.B we provide a rather detailed comment on the position state of the mass: “[...] the mass is prepared and kept in a superposition of coherent states centered around mean positions $x^{(i)}$ and zero momentum [...] Moreover, we take the mean positions $|x^{(i)}\rangle$ to be sufficiently far apart such that the overlap between the corresponding coherent states is negligible. This allows us to neglect the spread of the wavefunction in position space and effectively describe the massive object in terms of a superposition of position eigenstates $|x^{(i)}\rangle$.”*

We further neglect any small fluctuations of the momentum around zero and their contributions to the stress energy tensor, which we now clarified in the manuscript. This simplification is justified in the regime considered in our work but we agree that it would be an important direction for generalization that goes beyond the scope of this paper but that we hope to address in future work. We thank the referee for their observation.

Finally, regarding the comment on whether q_M should be an operator from the start: Since we are considering superpositions of semiclassical (coherent) states of the mass and neglecting the fluctuations, we regard it as valid to treat q_M in each branch as a classical number.

RC: *Reading this paper, I am coming to the conclusion that the QRF transformations apply only in the case of weak gravitational fields. In fact, it is completely unclear to me whet it would even mean to have a frame*

where the source of gravity is in a superposition if it is not possible to localise it in the first place, nor to give it a definite Hilbert space.

AR: *Again, we are unfortunately unsure where the confusion comes from in the statement “what it would even mean to have a frame where the source of gravity is in a superposition if it is not possible to localize it in the first place”. If it is about the position-momentum uncertainty, we would like to refer to our response directly above. Regarding the issue of defining the Hilbert space, see our response to the third comment.*

RC: *Finally, regarding the “covariance of dynamical laws under quantum coordinate transformations”, I do not understand how they could work (assuming they do) beyond the case where the source of gravity does not self interact, in contradiction with what we expect. In fact, In frame R the mass M is in superposition and we expect its state to decohere somehow. In frame M, it is in a definite state and it should not decohere. Thus, will it decohere or not? I cannot believe that the process of decoherence is frame dependent. This leads me to conclude, as noted above, that this work could be, at best, predictive for timescales shorter than those when self decoherence of a mass in a nonclassical begins to take place.*

AR: *An important point is made here: If the superposition decoheres in one frame, we would expect it to decohere in the other frame, too. In our work, we take it as given that if M is in a definite state (in the frame of M), it should not decohere, and from this conclude that any theory, such as gravitational collapse models, which assumes that it decoheres in the frame of R contradicts a basic symmetry principle. This issue is illustrated in Figure 5 and discussed in the Discussion section.*

RC: *Given the above, the claims of generality in the Discussion are over the top.*

RC: *The claim in “However, we go beyond the ... being related by a perturbative contribution.” Is incorrect. The ability to define a definite Hilbert space for the gravitating mass can be done, in the simple way that the authors implement their transformations, only in the linearised regime. Beyond that, I understand that it is even unclear how to univocally assign Hamiltonians to strongly gravitating and dynamical quantum systems, and to determine their position.*

AR: *We concede that it would be challenging to extrapolate the concrete construction of the Hilbert space in this work directly to the non-perturbative regime. However, this is a problem of formulating Hamiltonian formalism within classical spacetime background and has nothing to do with our argument. In fact, the generalized symmetry principles underlying our argument do extend to non-perturbative regimes. Indeed, we demonstrated this in recent work on treating superpositions of arbitrary conformally related metrics (see arXiv: 2207.00021 [quant-ph]).*

Overall comment

RC: *Given my comments above, it is clear that this work might be valid, if any, only when linearising general relativity and working to lowest order. In this sense, even if the authors were to reply to all of my criticisms and convince me that they are right, or fix the parts where the issues that I raise are indeed valid, it is not clear what statements about gravity are left. Spacetime, in this work, is Minkowski and the linearisation and first order regime means, de facto, that the authors are ignoring all of the interesting (e.g., nonlinear) aspects of gravity and working in a special relativistic setting with some Newtonian potentials where the sources of gravity do not have time to self-interact. While this can be interesting, and I think it is, it also means that the work is oversold. In addition, I would like to make the point that the authors cannot get away from this by just saying that one can easily extend their work to nonlinear and strong gravity, and that this will be done in later work. Either it is clear (theoretically and formally, not by words) already here*

that everything does carry through to strong gravity/nonlinear regimes, or the claim about adding insight on the overlap of general relativity and quantum mechanics loses support.

AR: *We hope that we have adequately addressed these issues in our responses above.*

References

RC: *The authors should add at least the following references which, I understand, are not directly relevant to the calculations but are for sure relevant for the broad context: [New Journal of Physics 21, 043047 (2019)] [Physics Letters B 54, 182-186 (2016)] I also take the opportunity to note that (some of) the authors, in the recent years, tend to ignore - for whatever reason - the work of colleagues (among others) that authored the papers suggested here. I would recommend the authors to be more “inclusive” (a buzzword invoked on a hourly basis nowadays) in the future.*

Final comment.

RC: *I am aware of the fact that my report is quite critical, and I do believe that this work is fundamentally flawed. Nevertheless, while I stand against publication, I am open to be proven wrong. I do not see how this could happen, but nevertheless I am can potentially change my opinion. I wished to make this clear with the authors, who might otherwise have the impression that I just do not like their work.*

AR: *Regarding the citation request, we agree with the referee that the two papers mentioned are not relevant to the calculations. The argument that they are relevant in a broader context can be applied to a multitude of papers in the broad and rapidly growing field at the interplay between gravity and quantum physics. In this regard, we have already made a selection of the works we deem most relevant and influential for our manuscript in the introduction.*

Response to Reviewer # 2

We would like to thank the referee for carefully studying our manuscript, inquiring about points that were unclear and pointing out possible modifications that make it more easily understandable for the reader. We hope that we provide an adequate response to the points raised in the report below.

RC: *(1) The prerequisite for using QRF reference frame transformation is that the mass configurations in different branches are related by relatively stable transformation. What is the relatively stable transformation between the mass configurations in different branches?*

AR: *The prerequisite for the existence of a QRF in which the mass configuration is definite is that configurations across the different branches (in the original frame) are related by transformations that preserve the relative coordinate distance between the constituents of the mass configuration. This amounts to global rotations and shifts. On the level of quantum states, this enters in Equation (41) through the action of the rotation and shift operators on the R and M subspaces, followed by the SWAP operator.*

RC: *(2) In the 4st paragraph of sec. I, the authors claimed that “The existence of test particles breaks the equivalence of the mass configurations related by these transformations in different branches”. I would like to know how this inequality is dealt with in the later calculations.*

AR: *Let us clarify that there are two types of equivalence discussed in the paper: the equivalence (or rather non-equivalence) of the configurations in the different branches of the superposition and the (postulated) equivalence of the descriptions of the entire quantum state in the different reference frames. The statement alluded to above refers to the first type of equivalence and aims to establish that we are considering an interesting scenario in which the configurations in the different branches are physically distinct from one another. This inequivalence thus simply characterizes the scenarios we are considering in the remainder of the paper and thus does not represent an issue that needs to be dealt with.*

RC: *(3) How to explain all the masses are in a definite position for all amplitudes in the bottom right and bottom left of Fig.4, and what does represent in the Fig.4?*

AR: *We are considering superpositions of semiclassical mass configurations. In each branch, we thus have a semiclassical configuration for which one can associate a definite position to each constituent of the configuration. This is reflected in Figure 4, in which the left and right subfigures depict the respective branches.*

RC: *(4) In the text before Eq. (32), the author applying the generalized transformation operator given in Eq. (33). I suggest that Eq. (33) be changed to Eq. (12).*

AR: *We changed this point in the manuscript.*

Reviewer comments:

Reviewer #1 (Remarks to the Author: Overall significance):

See attached PDF.

Reviewer #1 (Remarks to the Author: Impact):

See attached PDF.

Reviewer #1 (Remarks to the Author: Strength of the claims):

See attached PDF.

Reviewer #1 (Remarks to the Author: Reproducibility):

See attached PDF.

Reviewer #2 (Remarks to the Author: Overall significance):

The authors have properly replied my questions about the stable structure maintained by the components of the mass configuration and the theoretical tools utilized. However, they use a simplified model and many idealized assumptions in the present manuscript, which would shield a few realistic effects. I suggest the authors to state the effects that are ignored in the present model.

The authors have appropriately credited previous work and the claims in the present paper have appropriately discussed in the context of previous literature

REFeree REPORT

I have have been asked to review the reply to my report on the manuscript titled “Falling through masses in superposition...”.

I have started reading the reply but I had to interrupt it at a crucial point because I need to clarify some aspects about which I am very convinced the authors are mistaken. Without agreement on these aspects, I cannot continue reading the remainder of the reply.

I am focussing on the part that reads “RC *The authors say...*” and the reply “AR: *Firstly, let us note that...*”. In the following, I will try to reiterate my point, which I am certain that it is correct, and I hope to be clearer this time.

I start by thanking the referees for reminding me that quantum mechanics is linear. I had forgotten that.

Quantum field theory is indeed linear (in the sense of the linearity of quantum mechanics) as the authors suggest. Therefore, as the authors correctly point out, the different Feynman diagrams that are usually constructed to make sense of particle physics scattering processes interfere to give the final scattering amplitude. However, and this is a key difference with the work of the authors, and referencing to, say, Quantum Field Theory by M. Srednicki in its 2007 edition (you can find it for free on M Srednicki’s personal website), the derivation of the path integral approach and the Feynman diagram techniques requires at least Chapters 5 and 6, which have a quite long list of assumptions. It transpires, at least to me, that while path integral techniques are a fundamental tool for particle physics, it is not clear that they can be applied vis a vis to each and every scenario in physics. In particular, they are used in conjunction with the LSZ reduction formula, which crucially assumes that the incoming and outgoing excitations are **asymptotically noninteracting and well defined**. This means that far in the past and in the future the field excitations are well separated and therefore basically free. It is **only in this case** that the scattering process is understood, and thus the Feynman diagrams make sense. It is completely un-evident, therefore, that the scattering analogy would work in the paper under consideration beyond the lowest order \sqrt{GM} . In fact, the masses are not scattering and therefore asymptotically noninteracting and it is unclear (at least to me) therefore how to define the scattering matrix in the context of this work beyond lowest order \sqrt{GM} .

Second: let us keep using the pictorial representation of Feynman diagrams for the sake of argument, philosophically in the same fashion that the authors use the superposition of Earth locations in their work although the Earth cannot effectively be put in superposition.

The authors have computed a lowest order calculation in \sqrt{GM} . In quantum field theory, there are higher orders, an infinity of them, as the authors surely know. Thus, when considering the scattering processes mentioned above, these higher orders can and will contribute to the final answer. In standard high-energy physics calculations one can indeed ignore higher orders and focus on the lowest one if that precision is enough for his purposes. Since the probabilities computed thereof can be seen as statement regarding the eventuality of detecting certain well defined outputs in the (infinitely far) distant future given well defined inputs in the (infinitely far) distant past, the lowest order answer can be satisfactory. Now, in the present case the authors are not considering a scattering process. If they are, then this should be clear from the start. However, I understand that they are not, thus they cannot ignore the higher order corrections to the problem at all times. This is the core of my argument. The authors are **not** working in an analogous framework to scattering processes in high-energy physics like scenarios, and therefore their quantum field theory-like analogy is believable to **lowest order only** and **not for all times**. The authors **cannot** assume that the conclusions of their work do apply at all times, and for higher orders in the coupling constant. These are

severe limitations to their claims. On the other hand, if they would like to extend the claim for all times, and therefore making it a more universal result, they need to show that the assumptions and the scheme can indeed be predicting in such case. It is not enough to claim something as crucial as this for the work, and then cite previous work as support.

A quick note: What I argue is even true in standard quantum mechanical processes: if you cannot assume that the input and output excitations are well defined and non interacting in your far past and far future, then you cannot use the scattering-like approach, but you will have to study the problem for times that are short enough for the validity of whatever approximations you are using to apply. Some of the authors are world experts in the field of quantum physics and I am sure that they will recognise what I mean here.

Recommendation

I come to my recommendation. As I said before, I believe that research in this area is of current interest. However, the authors make statements and conclusions that are well beyond the scope and validity of their work. I cannot recommend this work for publication unless the authors do understand my point above and therefore implement the necessary changes that will be required.

Author Response to Reviews of

Falling through masses in superposition: quantum reference frames for indefinite metrics

A.-C. de la Hamette, V. Kabel, E. Castro-Ruiz, Č. Brukner
GUIDEDOA-22-00431A

Response to Reviewer #1

AR: *It is reassuring that the referee seems to agree with us on the linearity of Quantum Mechanics and QFT, as this seems to be at the heart of this exchange. In particular, linearity rules out any intrinsic decoherence due to the alleged “interaction between different branches” and the “gravitons exchanged between branches”, to refer to the referee’s formulation in the first report, in which this point is cited as the main criticism of our results. Indeed, in our previous reply, we made the remark on Feynman diagrams as an analogy to illustrate the linearity of QFT. We do not rely on perturbative methods in our work. We were solely using linearity to invalidate the referee’s claims that the branches ought to start interacting with each other. More generally, in the present work, we solely rely on the assumption that massive objects can, in principle, be placed in a linear superposition of two positions as well as the key postulate, the covariance of physical laws under quantum reference frame transformations. It therefore does not matter whether perturbative techniques, such as Feynman diagrams, apply to the situation considered here. Although one can expect that perturbation theory breaks down when the UV cutoff is of the order of magnitude of the Planck mass, we are not considering scenarios in which the branches of the superposition can be related to each other by a perturbation in the first place. Therefore, the referee’s arguments do not invalidate the claim that our arguments can in principle be extended beyond the weak-field regime.*

Response to Reviewer # 2

Overall significance

RC: *The authors have properly replied my questions about the stable structure maintained by the components of the mass configuration and the theoretical tools utilized. However, they use a simplified model and many idealized assumptions in the present manuscript, which would shield a few realistic effects. I suggest the authors to state the effects that are ignored in the present model.*

The authors have appropriately credited previous work and the claims in the present paper have appropriately discussed in the context of previous literature

AR: *We thank the referee for their suggestion; we have included an additional paragraph at the end of Sec. II.D, in which we discuss a few realistic effects that would occur in any experimental implementation: the gravitational field of the environment and the decoherence of the massive superposition. We reproduce here the relevant paragraph:*

Of course, any experiment that measures the gravitational effects of massive superpositions will face significant challenges. Firstly, one would need to suppress the gravitational and non-gravitational fields sourced by any object other than the gravitational source under consideration. By carefully controlling the gravitational field of the environment, such that there are no significant differences across the branches of the superposition, one should be able to isolate the effects sourced by the massive configuration. Furthermore, due to the interaction with an external environment, the gravitational source and the probe particle can become entangled with their surroundings and may lose their quantum coherence. Many specific models in which a system interacts with its environment have been studied including collisional and thermal decoherence [1, 2, 3] as well as the decoherence induced by the gravitational field background [4, 5, 6, 7]. Any experiment will have to be performed within a time frame shorter than the decoherence time.

References

- [1] K. Hornberger, J. E. Sipe, and M. Arndt, "Theory of decoherence in a matter wave talbot-lau interferometer," *Phys. Rev. A*, vol. 70, p. 053608, Nov 2004.
- [2] O. Romero-Isart, "Quantum superposition of massive objects and collapse models," *Phys. Rev. A*, vol. 84, p. 052121, Nov 2011.
- [3] M. Carlesso and A. Bassi, "Decoherence due to gravitational time dilation: Analysis of competing decoherence effects," *Physics Letters A*, vol. 380, no. 31, pp. 2354–2358, 2016.
- [4] T. Oniga and C. H.-T. Wang, "Quantum gravitational decoherence of light and matter," *Phys. Rev. D*, vol. 93, p. 044027, Feb 2016.
- [5] C. Anastopoulos and B. L. Hu, "A master equation for gravitational decoherence: probing the textures of spacetime," *Classical and Quantum Gravity*, vol. 30, p. 165007, Jul 2013.
- [6] B. Lamine, R. Hervé, A. Lambrecht, and S. Reynaud, "Ultimate decoherence border for matter-wave interferometry," *Phys. Rev. Lett.*, vol. 96, p. 050405, Feb 2006.
- [7] M. P. Blencowe, "Effective field theory approach to gravitationally induced decoherence," *Phys. Rev. Lett.*, vol. 111, p. 021302, Jul 2013.

Author Response to Reviews of

Falling through masses in superposition: quantum reference frames for indefinite metrics

A.-C. de la Hamette, V. Kabel, E. Castro-Ruiz, Č. Brukner
GUIDEDOA-22-00431C-Z

Response to Reviewer #1

We would like to thank the referee for their time and effort in reviewing our manuscript. In the present response, we would like to expand on the points raised in the last report and explain to what extent they apply to our results. In particular, we want to address the neglect of higher order terms and make precise why our paper is not based on this assumption.

To this end, let us begin by reviewing the perturbative approach to (quantum) gravity, following Chapters 1 and 2 of [1]. Generally, it is possible to expand the metric tensor $g_{\mu\nu}$ around a fixed metric. Often, this is done around the flat Minkowski metric and with the gravitational coupling $\kappa = \sqrt{32\pi G}$ (for $\hbar = c = 1$) as an expansion parameter, such that

$$g_{\mu\nu} = \eta_{\mu\nu} + \kappa h_{\mu\nu}, \quad (1)$$

where $h_{\mu\nu}$ is referred to as the metric perturbation. An expansion in the perturbation $\kappa h_{\mu\nu}$ and a neglect of higher order terms is only valid if the latter is “small” compared to the background metric. As a concrete example, take the Schwarzschild metric sourced by an object with mass M ,

$$g = - \left(1 - \frac{2GM}{r}\right) dt^2 + \left(1 - \frac{2GM}{r}\right)^{-1} dr^2 + r^2 g_\Omega, \quad (2)$$

where g_Ω denotes the metric on the two-sphere. Let us now see at which scales perturbation theory around the flat Minkowski metric is valid and when it breaks down. The time-time component of the Schwarzschild metric $g^{00} = \left(1 - \frac{2GM}{r}\right)$ can be seen as a perturbation around the time-time component of the Minkowski metric $g^{00} = 1$ if $\frac{M}{r} \ll \frac{1}{2G} = \frac{1}{2}M_{\text{Pl}}^2$ where M_{Pl} denotes the Planck mass. Thus, when the ratio of the mass M of and the distance r from the gravitational source approaches $\frac{1}{2}M_{\text{Pl}}^2$, the perturbative contribution grows too large and perturbation theory loses its validity. Importantly, this means that perturbation theory breaks down exactly at the Schwarzschild radius $r_s = 2GM$ as this is the distance at which $M/r_s = 1/(2G)$ for any massive object. However, already outside of the Schwarzschild radius but in its vicinity, perturbation theory would no longer provide a good approximation to the full theory as the higher order terms contribute significantly to the expansion.

More generally, in a perturbative approach to quantum gravity, one can view $h_{\mu\nu}$ as a field on flat spacetime. In this case, an expansion in the coupling constant κ will be valid only at certain energy scales. In particular, κ has dimension $\sqrt{G} = 1/M_{\text{Pl}}$. Thus, this perturbative approach to quantum gravity is valid only for energy scales smaller than $M_{\text{Pl}}c^2 \approx 10^{18}\text{GeV}$. In other words, one can neglect terms of higher order in κ only as long as the energy scale is below the Planck mass.

Given the prevalence of perturbative approaches to scenarios in quantum gravity, it is thus natural to be alarmed whenever energy scales larger than the Planck energy or regimes in which $\frac{M}{r} \gtrsim \frac{1}{2G} = \frac{1}{2}M_{\text{Pl}}^2$ are considered. In that case, the neglect of higher order terms would be extremely problematic as their corrections grow with increasing order.

Fortunately, the approach devised in our work does not rely on perturbative methods at any point. By restricting the set of configurations that we treat to superpositions of mass configurations related by global translations and rotations, we are able to make predictions, solely based on the assumption of covariance under quantum reference frame transformations. This assumption allows us to map a situation in which we have a gravitational source in a superposition of two or more locations to one in which it is in a well-localized position. We thus avoid any mathematical treatment of the “quantum metric” sourced by such a configuration. On the contrary, we can use classical general relativity to determine the spacetime geometry and consequently the motion of test particles and clocks. Therefore, we only rely on the regime of validity of full general relativity. In particular, our model makes predictions for realms in which perturbation theory breaks down. That is, if the metrics sourced by the gravitating object in each of the branches separately cannot be obtained by perturbations around the *same* background, our approach can make predictions whereas perturbation theory cannot be applied. Concretely, consider a probe particle in the presence of a black hole in a superposition of two distant locations – one close to and one far away from the probe. If we wanted to apply linearized quantum gravity, we would have to describe the gravitational field at the location of the probe particle as a superposition of perturbations around the *same* spacetime background g^0 . However, if the superposition of the black hole is large enough, there does not exist a g^0 such that

$$g^1(x_p) = g_{\mu\nu}^0(x_p) + h_{\mu\nu}^1(x_p), \quad (3)$$

$$g^2(x_p) = g_{\mu\nu}^0(x_p) + h_{\mu\nu}^2(x_p), \quad (4)$$

where $h^1, h^2 \ll g^0$. Our approach does allow to change to a quantum frame in which the black hole is localized at one definite position, thus sourcing a classical gravitational field in which the dynamics of the probe particle can be computed. Outside these regimes, when perturbation theory is valid, it is possible to *compare* our results with the predictions of perturbative approaches. However, our method does not rely at any point on a perturbative expansion. Thus, there is no neglect of higher order terms in our approach.

Note that it is not the goal of this approach to construct a full theory of quantum gravity. Rather, it aims at treating specific scenarios at the interface between quantum theory and gravity that can be tackled using extended symmetry principles. In this sense, it is *complementary* to full quantum gravity approaches such as loop quantum gravity as well as perturbative approaches to quantum gravity.

As an additional clarification for the reader, we added the above explanations in a slightly more detailed way in the supplementary material, see “Supplementary Note D : Comparison with perturbative approaches to quantum gravity.”.

Finally, we want to briefly explain why we did not expand more on the topic on scattering processes in our latest reply to Referee #1. In their second report, the referee focused strongly on issues related to scattering processes, which, as they pointed out themselves, do not apply to any scenarios considered in our work. Specifically, scattering requires “asymptotically noninteracting and well defined” in- and out-states, a condition not met by any situation we consider. The reason this discussion started in the first place was an analogy we made in our first reply to Referee #1 in order to illustrate the linearity of quantum field theory. As we hope to have made clear in this letter, however, we do not at any point employ the perturbative methods used to compute scattering amplitudes in quantum field theory.

We have endeavored to fully address Referee #1’s concern regarding the neglect of higher order terms and hope to have clarified that it does not concern the validity of our results.

References

- [1] M. Maggiore, *Gravitational Waves. Vol. 1: Theory and Experiments*. Oxford Master Series in Physics, Oxford University Press, 2007.

REVIEWERS' COMMENTS:

Reviewer #1 (Remarks to the Author: Overall significance):

The authors claim to use quantum reference frame (QRF) transformations to compute dynamics of objects in the presence of indefinite spacetime metrics. The key to their work is the fact that, under certain assumptions, they can perform a transformation where the source of the indefinite metric becomes well defined and therefore computations reduce in complexity the calculation of dynamics. Applications are discussed.

Reviewer #1 (Remarks to the Author: Impact):

I think this work is interesting, but it does not warrant publication in Nature Comms. or Nature Phys. I think a more technical journal would be suited.

Reviewer #1 (Remarks to the Author: Strength of the claims):

As discussed already many times, this work is interesting and certainly warrants publication in some form.

The authors have gone to great length to answer my specific criticisms, which I acknowledge and appreciate.

Nevertheless, I remain adamant that the claim of applicability do not apply, and that this work has limited scope within a very restricted class of scenarios, for which self interaction must be excluded. This occurs only for a fixed amount of time as I have discussed before.

The authors have not proven that this is incorrect, because they ARE using - in their useful frame - low energy physics. Dynamics of low energy physics coupled to quantum superpositions must break down at a certain timescale yet to be determined. The dynamics they propose cannot be trusted beyond such timescale.

For these reasons, I recommend publication in a more technical journal from the Nature portfolio but I definitely recommend against the higher impact journals such as Nat. Comm or Nat. Phys.

Reviewer #1 (Remarks to the Author: Reproducibility):

Not applicable.